# Huntingtin's spherical solenoid structure enables polyglutamine tract-dependent modulation of its structure and function

Ravi Vijayvargia[1,2†], Raquel Epand[3], Alexander Leitner[4], Tae-Yang Jung[5,6,7], Baehyun Shin[1,2], Roy Jung[1,2], Alejandro Lloret[1,2‡], Randy Singh Atwal[1,2], Hyeongseok Lee[5], Jong-Min Lee[1,2], Ruedi Aebersold[4,8], Hans Hebert[6,7], Ji-Joon Song[5*], Ihn Sik Seong[1,2*]

[1]Center for Human Genetic Research, Massachusetts General Hospital, Boston, United States; [2]Department of Neurology, Harvard Medical School, Boston, United States; [3]Biochemistry and Biomedical Sciences, McMaster University, Hamilton, Canada; [4]Department of Biology, Institute of Molecular Systems Biology, Eidgenössische Technische Hochschule Zürich, Zurich, Switzerland; [5]Department of Biological Sciences, Cancer Metastasis Control Center, KAIST Institute for the BioCentury, Korea Advanced Institute of Science and Technology, Daejeon, Republic of Korea; [6]Department of Biosciences and Nutrition, Karolinska Institute, Solna, Sweden; [7]School of Technology and Health, KTH Royal Institute of Technology, Novum, Sweden; [8]Faculty of Science, University of Zurich, Zurich, Switzerland

*For correspondence: songj@ kaist.ac.kr (JJS); iseong@mgh. harvard.edu (ISS)

Present address: [†]Department of Biochemistry, The Maharaja Sayajirao University of Baroda, Vadodara, India; [‡]Facultad de Medicina, Universidad Autónoma de Querétaro, Santiago de Querétaro, Mexico

Competing interests: The authors declare that no competing interests exist.

**Abstract** The polyglutamine expansion in huntingtin protein causes Huntington's disease. Here, we investigated structural and biochemical properties of huntingtin and the effect of the polyglutamine expansion using various biophysical experiments including circular dichroism, single-particle electron microscopy and cross-linking mass spectrometry. Huntingtin is likely composed of five distinct domains and adopts a spherical $\alpha$-helical solenoid where the amino-terminal and carboxyl-terminal regions fold to contain a circumscribed central cavity. Interestingly, we showed that the polyglutamine expansion increases $\alpha$-helical properties of huntingtin and affects the intramolecular interactions among the domains. Our work delineates the structural characteristics of full-length huntingtin, which are affected by the polyglutamine expansion, and provides an elegant solution to the apparent conundrum of how the extreme amino-terminal polyglutamine tract confers a novel property on huntingtin, causing the disease.

## Introduction

Huntingtin is the entire large protein product (>350 kDa MW) of the *Huntingtin* gene (*HTT* previously *HD*). Huntingtin has a segment of polyglutamine near its amino terminus (Amino-terminal) that is encoded by a polymorphic CAG trinucleotide repeat. If expanded above 38-residues, this mutation causes Huntington's disease (HD), a dominant neurodegenerative disorder (*Huntington's Disease Collaborative Research Group, 1993*). The strong correlation between the size of the expanded repeat and the age at diagnosis of HD motor, cognitive and psychiatric symptoms shows that CAG repeat-size is the primary determinant of the rate of the disease progression (*Brinkman et al., 1997*; *Snell et al., 1993*). This biological relationship also provides a human patient-based rationale for delineating the HD disease-trigger in studies with an allelic series

**eLife digest** Huntington's disease is an inherited disorder that occurs in adulthood and sometimes in children. It causes progressive damage to the brain and people with the condition develop memory loss, movement difficulties, confusion, and other symptoms of mental decline. Eventually, the disease leads to death. Mutations in the gene that encodes a protein called huntingtin cause Huntington's disease. Individuals who inherit just one copy of the mutated gene develop the condition. No treatments currently exist that can slow or stop disease progression.

Genetic and molecular studies are beginning to shed light on how mutations in the gene encoding huntingtin cause the disease. Normally, the protein has a section near its tail end made up of the amino acid glutamine repeated around 23 times. Mutations that increase the number of glutamines to more than 38 cause Huntington's disease. The more extra glutamines there are in this region of the protein, the earlier in life the disease symptoms begin. But it was not clear how these extra glutamines near the tail of huntingtin affect the structure and behavior of a protein that is more than 3,000 amino acids long.

Now, Vijayvargia et al. have revealed why the tail end of huntingtin is so important. Several biophysical methods were used to determine the three-dimensional structure of the huntingtin protein. These methods revealed that the protein folds up into a hollow sphere and that its tail end is able to interact with the entire length of the protein and physically touches its opposite end.

To see this in more detail, Vijayvargia et al. used another experimental technique called crosslinking mass spectrometry to confirm which parts of the huntingtin protein are in close contact with each other. Together with the structural data, these experiments suggest that the stretch of glutamines is in the position to bring about subtle, but widespread, changes throughout the huntingtin protein. That is to say, that having more glutamines slightly changes the curve of the sphere and alters the way different parts of the protein interact.

Together the new findings explain why mutations that alter the tail of huntingtin affect the rest of the protein. Further work will now aim to provide a more-detailed structure of the huntingtin protein and to investigate what other roles of huntingtin are affected by the increased number of glutamines in the protein's tail. These insights may help scientists understand how the mutated protein causes brain decline.

designed to determine the effects of progressively increasing the size of the mutation. Consistent with genetic studies in distinct CAG-expansion neurodegenerative disorders and CAG knock-in mice that replicate the HD mutation, the mechanism that triggers the disease process that leads to the characteristic vulnerability of striatal neurons in HD is thought to involve a novel gain of function that is conferred on mutant huntingtin by the expanded polyglutamine segment (*Gusella and MacDonald, 2000*; *Nucifora et al., 2001*; *Trettel et al., 2000*).

By analogy with other members of the HEAT/HEAT-like (Huntingtin, Elongation factor 3, protein phosphatase 2A, Target of rapamycin 1) repeat family (*Andrade and Bork, 1995*; *Perry and Kleckner, 2003*), huntingtin is likely a HEAT domain solenoid that functions as a mechanical scaffold for multi-member complexes (*Grinthal et al., 2010*; *Takano and Gusella, 2002*). Huntingtin's large size and predicted predominant HEAT/HEAT-like repeat domain structure is well conserved through 500 million years of evolution (*Seong et al., 2010*). The polyglutamine region is not conserved in some huntingtin orthologues (*Seong et al., 2010*), implying a role as an extra feature that fine-tunes huntingtin structure and function. Indeed, testing this idea, we have previously demonstrated, with purified recombinant human huntingtins in a cell-free assay, that lengthening the polyglutamine tract quantitatively enhances the basal function of huntingtin in stimulating Polycomb repressive complex 2 (PRC2) histone methyltransferase (*Seong et al., 2010*).

The structures of smaller HEAT/HEAT-like repeat solenoid scaffold proteins, such as PR65/A and Importin β, have been solved to high-resolution, and each has been shown to assume a distinctive extended curvilinear shape determined by the specific stacking characteristics of its HEAT/HEAT-like repeats (*Cingolani et al., 1999*; *Groves et al., 1999*). The topology of the huntingtin solenoid is expected to reflect the specific stacking characteristics of α-helical HEAT/HEAT-like repeats that

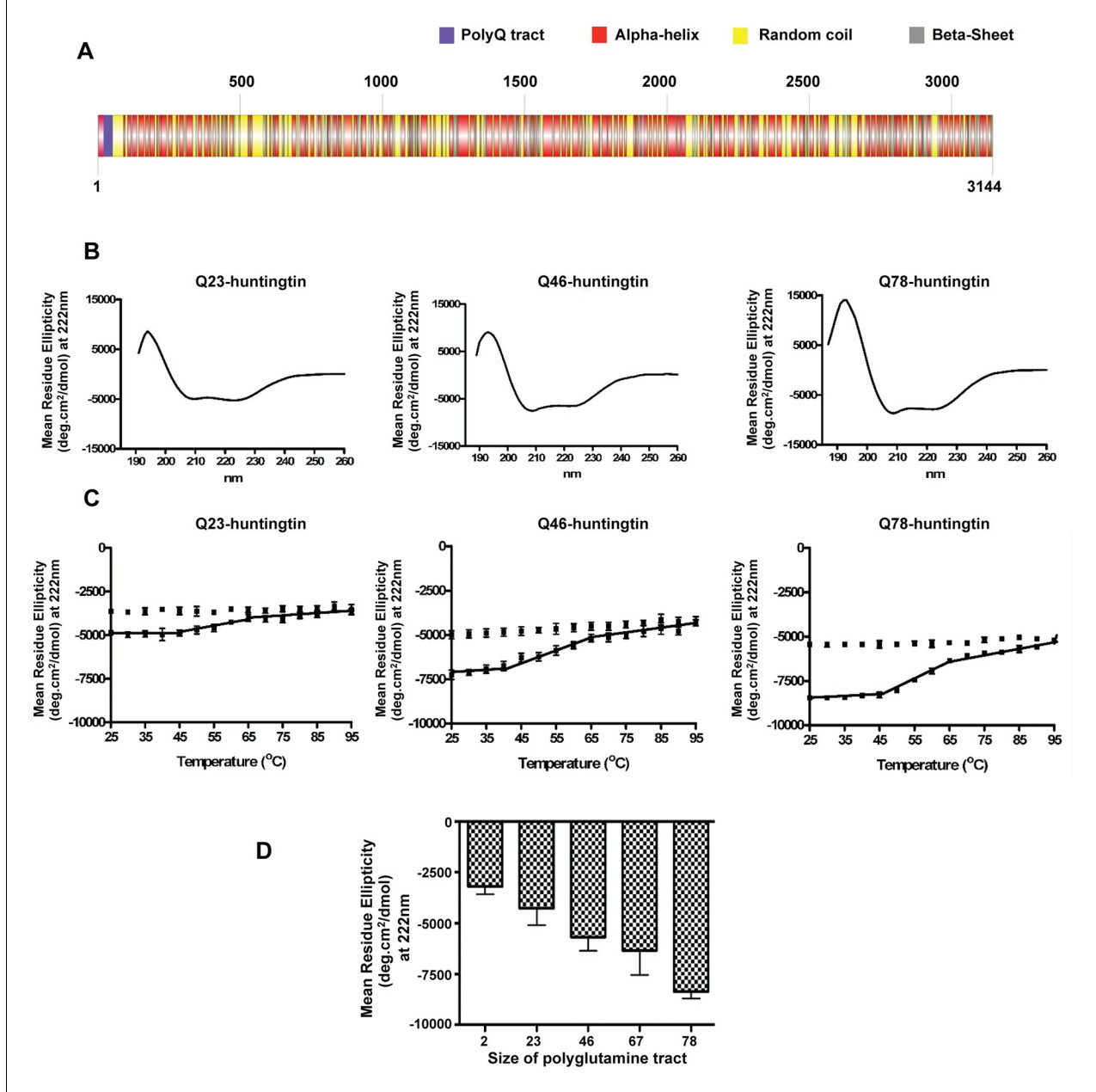

**Figure 1.** Huntingtin secondary structure is modulated by the length of polyglutamine tract. (A) Human huntingtin amino acid sequence (Homo sapiens; NP_002102) was analyzed for predicted secondary structure using: NORSp (*Liu and Rost, 2003*) and PROF (Profile network prediction Heidelberg) (*Rost et al.,1994*). Stick model of human huntingtin protein (3144 amino acids) was generated depicting the predicted alpha helical (red), random coil (yellow) and beta sheet (grey) regions. The polyglutamine tract in the amino-terminus is indicated in purple. (B) The far UV-wavelength scan at 25°C of these purified huntingtin proteins generates a curve typical of α-helical proteins. (C) Thermal behavior of Q23-, Q46- and Q78-huntingtin. The heat denaturation curves, from 25 to 95°C, of all proteins showed the similar pattern of irreversible thermal denaturation starting their denaturation above 40°C by MRE values at 222 nm. Due to inherent variation caused by inefficient mixing in the cuvette with taking readings every five degrees of heating, their heat denaturation curves were acquired in duplicates. Solid line represents heating to 95°C; dotted line represents cooling from 95°C. (D) An average (MRE) in units of deg.cm2/dmol, at 222 nm wavelength characteristic of an α-helix), from two independent experiments, was plotted against the length of the polyglutamine tract of huntingtin proteins (bars represent mean ± SEM). Temperature was 25°C.

The following figure supplement is available for figure 1:

**Figure supplement 1.** Normalization of purified huntingtin proteins with varied polyglutamine tract length.

span the molecule. The shape imparted by intramolecular stacking cannot be predicted because of the idiosyncratic nature of HEAT/HEAT-like repeats, which are loosely conserved ~34 amino acid bipartite α-helical units (*Takano and Gusella, 2002*). Nevertheless this shape must enable modulation by the amino-terminal polyglutamine segment. It seems reasonable that this may involve some structural feature that is critical to huntingtin function. One possibility is structure-dependent post-translational modification. Human huntingtin is phosphorylated, at more than seventy modified serine, threonine and tyrosine residues (*Hornbeck et al., 2012*). Indeed, a comparison of lines of transgenic modified *HTT* BAC mice has implicated unique amino-terminal serine phosphorylation in protection against deleterious effects of mutant huntingtin (*Gu et al., 2009*) and the polyglutamine expansion at the amino-terminal causes a trend of hypo-phosphorylation in all sites, including sites near the carboxyl-terminus (*Anne et al., 2007*; *Schilling et al., 2006*; *Warby et al., 2005*), indirectly implying a long-range impact of the amino-terminal region on huntingtin structure and function.

In order to solve the apparent puzzle of how huntingtin's solenoid structure may enable quantitative or qualitative (or both) modulation of huntingtin function, according to the size of the amino-terminal polyglutamine tract, we have extended initial observations showing a likely flexible α-helical structure by conducting systematic biophysical and biochemical analyses of members of a panel of highly purified human recombinant huntingtins, with varied lengths of polyglutamine tracts (*Fodale et al., 2014*; *Huang et al., 2015*; *Li et al., 2006*).

## Results

### Huntingtin α-helical structure is quantitatively altered with polyglutamine tract size

It has been reported previously that purified huntingtin exhibits a predominantly α-helical secondary structure but among studies the impact of polyglutamine size has been inconsistent (*Fodale et al., 2014*; *Huang et al., 2015*; *Li et al., 2006*). To carry out a standardized evaluation, we performed circular dichroism (CD) analysis of a series of recombinant human huntingtins with different polyglutamine tract lengths (Q2-, Q23-, Q46-, Q67-, Q78-huntingtin, respectively) purified to homogeneity (*Figure 1—figure supplement 1*). The CD spectra (*Figure 1B*) of all of the huntingtins are consistent with a predominant α-helical secondary structure (*Liu and Rost, 2003*; *Rost et al., 1994*) (*Figure 1A*), with typical minima at 222 and 208 nm and a positive peak at 195 nm, and all exhibited the same irreversible thermal denaturation pattern, with secondary structure stable up to ~38–40°C, a gradual slow denaturation as the temperature is increased to 65–70°C, followed by aggregation and some precipitation (*Figure 1C*). These results imply the same basic core structure and stability regardless of the size of the expanded polyglutamine segment. Plotting an average of the Mean Residue Ellipticity (MRE) at 222 nm (characteristic of an α-helix) reveals an incremental quantitative effect of lengthening the polyglutamine tract at the amino terminus on the α-helicity of the entire molecule (*Figure 1D*).

### 3D EM analysis reveals a spherical shape with a central cavity and overlying Amino-terminus

We then investigated the proposal that huntingtin's shape may enable a structural impact of the amino-terminal polyglutamine tract, by performing single-particle electron microscopy (EM) of recombinant human huntingtins with polyglutamine tract lengths of 23- and 78-residues. These were purified to high homogeneity using a gradient purification method with mild crosslinking (GraFix) (*Kastner et al., 2008*), collecting only the monomer fraction for analysis to eliminate potential contributions from oligomeric structures that would confound interpretation of the results (*Figure 2—figure supplement 1*). Negative-stained micrographs of these proteins confirmed that the samples were highly homogeneous (*Figure 2—figure supplement 2*). A total of 10,169 particles were chosen for generating 2D class averages and 30 class averages were used for reconstructing a 3D EM map of Q23-huntingtin (*Figure 2—figure supplement 3A*). The EM map of Q23-huntingtin at about 30 Å resolution, estimated from Fourier shell correlation analysis, shows that the molecule adopts an overall spherical shape with 130 Å height and 100 Å width (*Figure 2A* and *Figure 2—figure supplement 4*). The overall shape of huntingtin was not apparently affected either through crosslinking or by the tag as 2D class averages of huntingtin with no cross linker or without tag also showed a similar

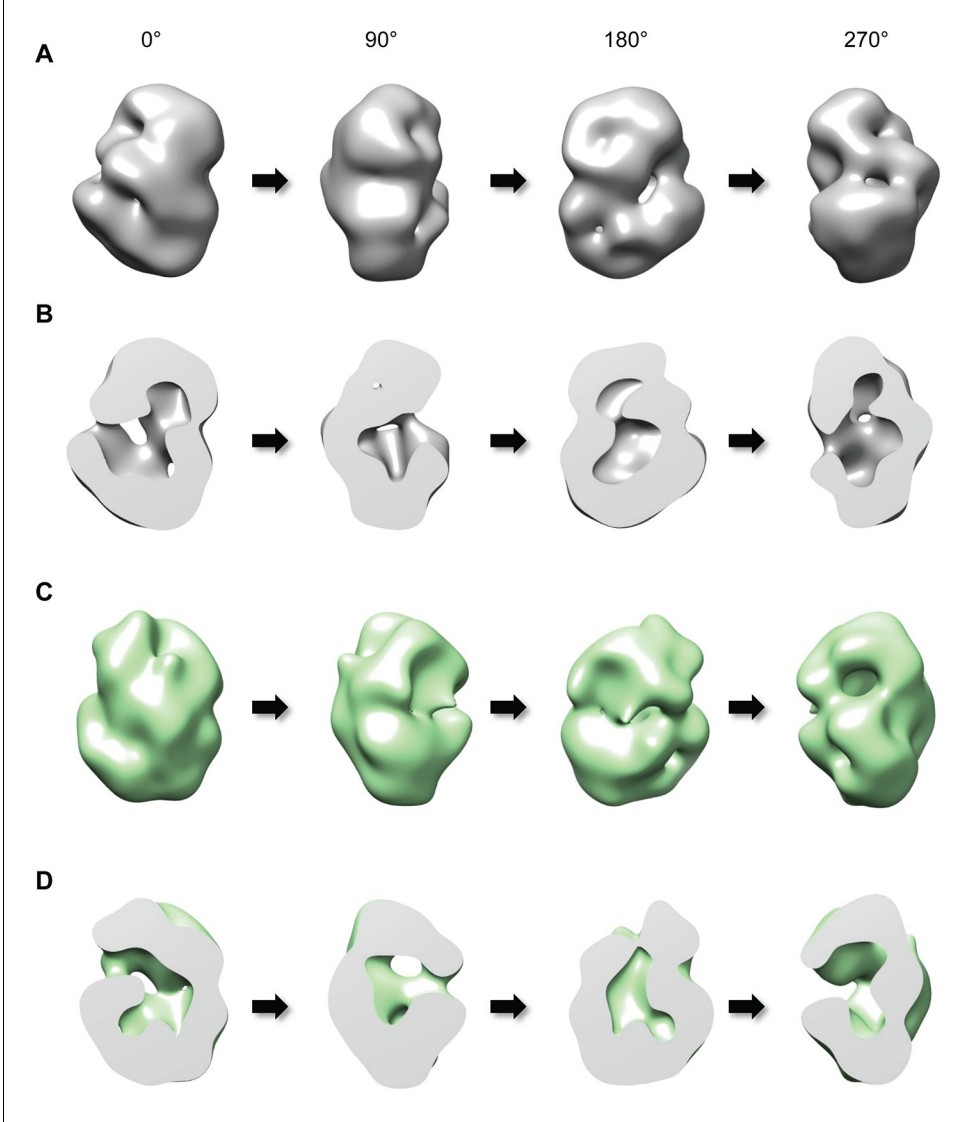

**Figure 2.** Three-dimensional reconstruction of negatively-stained Q23- and Q78-huntingtin. (**A**) 3D EM map of Q23-huntingtin was reconstructed from negatively-stained particles of Q23 monomer separated by GraFix. The resolution was estimated as 33.5 Å at 0.5 FSC. 3D map of Q23-huntingtin is shown in different orientation rotated about the y axis (0°, 90°, 180°, 270°). (**B**) Sectioned view of 3D EM map of Q23-huntingtin in the same orientation as in **A** revealing a large cavity inside of Q23-huntingtin. (**C**) 3D EM map of Q78-huntingin (32.0 Å at 0.5 FSC) was reconstructed as for Q23-huntingtin and shown in different angles rotated about the y axis (0°, 90°, 180°, 270°). (**D**) Sectioned view of 3D EM map of Q78-huntingtin in the same orientation as in **C** also showing a large cavity inside of Q78-huntingtin. This figure has additional supplement files: Figure supplement 1, 2, 3, 4, and 5

The following figure supplements are available for figure 2:

**Figure supplement 1.** Purification of huntingtin by GraFix.

**Figure supplement 2.** Negatively-stained micrographs of huntingtin at 50,000X magnification.

**Figure supplement 3.** 2D Class averages of Q23- and Q78-huntingtin.

**Figure supplement 4.** Fourier Shell Correlation (FSC) curves.

**Figure supplement 5.** Superimposition between 3D EM maps of Q23 and Q78-huntingtin.

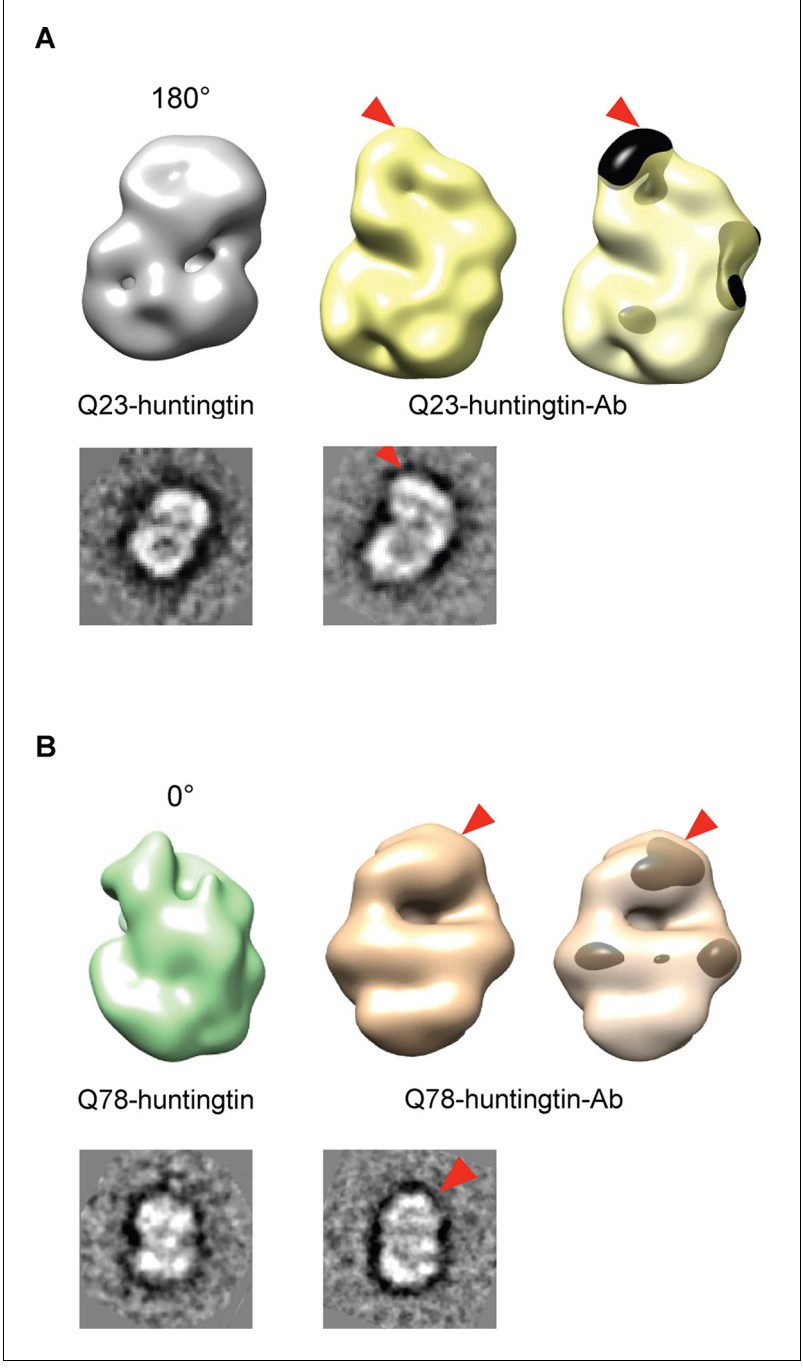

**Figure 3.** Detection of the amino-terminus region of huntingtin by electron microscopy. 3D reconstitutions of Q23-huntingtin and Q23-huntingtin antibody complex (**A**) or Q78-huntingtin and Q78-huntingtin antibody complex (**B**) are shown in gray and yellow or in green and brown, respectively. 2D class averages corresponding to each huntingtin and its antibody complex are shown below the 3D reconstituted EM map and the extra density is marked with a red triangle. The extra-density indicating antibody on 3D EM map is colored in black (**A**) or dark brown (**B**) with red triangles.

The following figure supplements are available for figure 3:

**Figure supplement 1.** Producing huntingtin-FLAG-Aantibody complex.

**Figure supplement 2.** Negatively-stained micrographs of huntingtin-antibody complexes at 50,000X magnification.

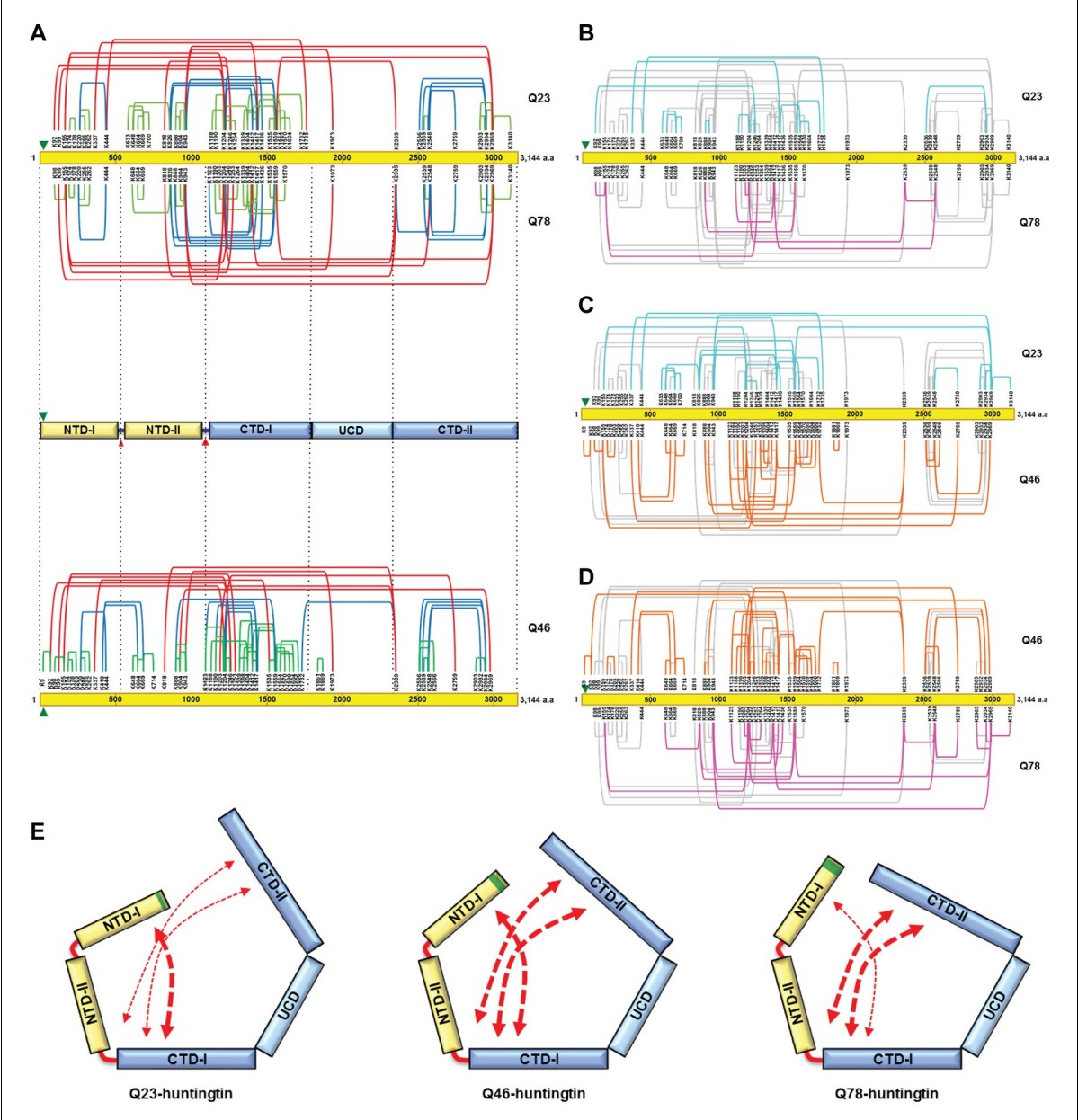

**Figure 4.** Cross-linking mass spectrometry analysis shows the intra-molecular interactions of Q23-, Q78- and Q46-huntingtin. The 3,144 amino acid primary huntingtin sequence (by convention Q23-huntingtin) is depicted as a yellow bar with the location of the polyglutamine tract indicated by the green arrowhead (**A**). The short-, mid- and long-range Lys-Lys cross-links by DSS identified in Q23-huntingtin (above the bar, Q23) and Q78-huntingtin (below the bar, Q78) by XL-MS are depicted by the green, blue and red-colored lines, respectively. Below that is a schematic view of huntingtin with five sub-domains delineated by the shared patterns of intra-molecular interactions; two amino-terminal (NTD-I, NTD-II) and three carboxyl-terminal (CTD-I, UCD and CTD-II), as defined relative to the landmark major protease-sensitive site at ~ residue 1200 identified previously (*Seong et al., 2010*), which is denoted by the large red arrowhead, while the secondary minor cleavage site is denoted by the small red arrowhead. The cross-links of Q46 huntingtin (Q46) identified in XL-MS analysis are also shown under the five sub-domains schematic. Lys-Lys cross-links by DSS unique to Q23-huntingtin, Q78-huntingtin and Q46-huntingtin are shown in cyan, pink, and orange, respectively in each pair-wise comparison of Q23 vs Q78 (**B**), Q23 vs Q46 (**C**) or Q46 vs Q78 (**D**). The amino-terminal (yellow) and carboxyl-terminal (blue) sub-domains are depicted in cartoons (**E**) to show more substantial interactions (red thicker dashed arrows) between NTD-I and CTD-I in Q23-huntingtin (left) and between CTD-II and NTD-II or CTD-I in Q78-huntingtin (right) and throughout both amino- and carboxyl-terminal sub-domains in Q46-huntingtin (middle). In all three huntingtins (all red dashed arrows), NTD-I folds to contact CTD-I and CTD-II also contacts CTD-I as well as NTD-II, implying that the physical impact of the polyglutamine tract at the very amino-terminal end has the opportunity to subtly but globally alter the entire huntingtin structure and function in a polyglutamine length-dependent manner.

*Figure 4 continued on next page*

*Figure 4 continued*

The following figure supplement is available for figure 4:

**Figure supplement 1.** Purification of huntingtin by ultracentrifugation after DSS cross-linking.

spherical shape (*Figure 2—figure supplement 3*). The outer volume of the structure can be roughly estimated as 861,829 Å$^3$ and a Mathew's coefficient (V$_M$) of 2.48, assuming that Q23-huntingtin is a sphere with 115 Å diameter with 348 kDa molecular weight. Considering the V$_M$=1.23 for the protein itself, Q23-huntingtin appears to contain a large solvent cavity (up to 50% by volume). Consistent with this estimation, the 3D EM reconstruction of Q23-huntingtin shows a large cavity in the core (*Figure 2B*). The analysis of negatively-stained Q78-huntingtin (*Figure 2C* and *Figure 2—figure supplement 3B*) disclosed a 3D map showing a similar overall spherical shape, with a large cavity in the core (*Figure 2D*). Although 3D maps were reconstructed de novo without other experimental methods such as random conical tilt, the high similarity of the shapes between Q23- and Q78-huntingtin 3D maps attests that huntingtin has the spherical structure with a cavity (*Figure 2—figure supplement 5*). Notably, manual superimposition of the 3D maps of Q23-huntingtin and Q78-huntingtin reveals potential differences in the two structures (*Figure 2—figure supplement 5*), which may reflect technical variation (image processing, stain distribution, sample heterogeneity), in addition to the structural effects of the lengthened polyglutamine segment that were foreshadowed by the altered CD spectra (*Figure 1D*).

We then attempted to locate the amino-terminus (17 residues adjacent to the polyglutamine tract) of huntingtin in the EM maps, by collecting images of negatively-stained purified complexes of antibody-bound amino-terminal FLAG-tags of the Q23- and Q78-huntingtin (*Figure 3* and *Figure 3—figure supplement 1* and *2*). We also proceeded to reconstitute 3D structure of Q23- and Q78-huntingtin-antibody complexes. Comparisons of the 2D class averages and 3D reconstituted structures between huntingtin alone and the huntingtin-FLAG-antibody complex pairs clearly reveals an extra density at the top of the structure (in the view shown) for both Q23- and Q78-huntingtin (*Figure 3*). These observations strongly imply that the extreme N-terminus, and by inference the adjacent polyglutamine tract, is folded back, forming a spherically shaped solenoid with an internal cavity, but is accessible at the outside surface, regardless of its length.

## Cross-linking-MS analysis reveals a modulated network of intramolecular contacts

To further examine structural characteristics such as folding, we then assessed intramolecular interactions within huntingtin as estimated from the spatial proximity of lysine residues in sucrose-gradient-selected monomeric Q23- and Q78-huntingtin, determined by disuccinimidyl suberate (DSS) cross-linking mass spectrometry (DSS XL-MS) analysis (*Leitner et al., 2014*) (*Figure 4—figure supplement 1*, *Supplementary file 1*). Based upon the spacing between the DSS-cross-linked lysine residues in the primary sequence, the interactions that were detected for either huntingtin can be grouped into three categories, depicted in *Figure 4A*. These comprise: #1) short-range interactions (within a 200 amino acid interval), which seem likely to occur within the same secondary structure element, including the contacts between pairs of adjacent lysines (e.g. K174-K178 and K664-K669), and the interactions between K220, K255 and K262; #2) mid-range interactions (201 to 1000 amino acid interval), such as K826-K1559, K943-K1559 and K2548-K2934; and #3) long-range contacts (interval of >1000 amino acids), including between the carboxyl-terminal K2969 and amino-terminal K943 residue. Inspection of the depiction of the short-range cross-link contact sites relative to the location of the protease-sensitive sites (*Seong et al., 2010*) indicates that huntingtin is likely to be composed of five distinct domains (*Figure 4A* upper panel). The location of the protease-sensitive major hinge region located at residues 1184–1254 defines the 150 kDa amino-terminal domain (NTD) and the 200 kDa carboxyl-terminal domain (CTD). A minor-protease sensitive site located at ~ residue 500 demarcates the NTD into NTD-1 and NTD-II. Interestingly, a region centrally located within the CTD showed strikingly few crosslinks, despite the presence of several lysine residues, strongly implying a distinct sub-domain that we call 'uncrosslinked' sub-domain (UCD), which, given the paucity of short range intramolecular contacts, may adopt a largely unfolded structure. The UCD is flanked by regions with numerous intramolecular contacts; the CTD-I in proximity to the major proteolysis site and the

carboxyl-terminal CTD-II (*Figure 4A*). Consistent with the hypothesis of five discernable huntingtin sub-domains, the results of hydrophobicity analysis show a transition in hydrophobicity prediction at the edge of each sub-domain (data not shown).

The mid- and long-range interactions occur between these sub-domains in a pattern that indicates close-proximity of the extreme amino- and carboxyl-terminal sub-domains. Specifically, NTD-I mainly interacts with CTD-I, while CTD-II interacts with NTD-II and notably with CTD-1. Thus, the pattern of mid- and long-range contacts supports a view of huntingtin folding such that the extreme amino-terminal subdomain (NTD-I), with its polyglutamine tract, and the extreme carboxyl-terminal subdomain (CTD-II) are close to each other by virtue of contacts that each makes with the NTD-II and CTD-I sub-domains that flank the major cleavage site.

Notably, the overall contact-patterns for the Q23- and Q78-huntingtin were similar, supporting observations of a generally similar core-stability (*Figure 1*) and shape (*Figures 2* and *3*). However, subtraction of the 38 crosslinks common to both proteins highlight networks of contacts that are relatively specific for either Q23-huntingtin (13 crosslinks) or Q78-huntingtin (8 crosslinks), as depicted in *Figure 4B*. To further examine the patterns of the internal interaction depending on its polyglutamine length, we also performed XL-MS analysis of Q46-huntingtin. First, the overall contact-patterns of Q46-huntingtin were similar and consistent with the five distinct domains (*Figure 4A* lower panel). Compared with those of Q23- and Q78-huntingtin, the unique contacts of Q46 reveal widespread interactions across the entire region of the protein (*Figure 4C and D*). These unique, largely mid- and long-range contacts disclose that Q23-huntingtin exhibits more unique interactions of the NTD-I with the CTD-I, and Q78-huntingtin displays more unique contacts between the CTD-II and the CTD-I and on occasion with NTD-II. On the other hand, the unique crosslinks of Q46-huntingtin reveal that CTD-I seems to interact with both NTD-I and CTD-II as if Q46-huntingtin posits an intermediate conformation between Q23- and Q78-huntingtin. (*Figure 4E*). These observations imply a subtle but detectable 3-dimensional structural impact of polyglutamine tract length as it increases.

## Discussion

We applied a systematic structure-function approach to delineate the features of huntingtin that conspire with its polyglutamine tract to comprise, in conjunction with some as yet unknown target, the dominant gain of function mechanism that triggers the pathogenic process in patients with HD. Our biophysical analyses of an allelic series of native recombinant human huntingtins now provide a satisfying solution to the mystery of how the amino-terminal polyglutamine tract may be in a position to modulate huntingtin structure and function. The results of EM and XL-MS analyses provide coherent support for a HEAT/HEAT-like repeat solenoid comprising a major hinge that delimits two large nearly equal-sized domains that fold such that the ends of each arm are in close proximity and the whole circumscribes an extensive internal cavity. Other HEAT repeat proteins such as nuclear importin and exportins have functional protein-protein binding interfaces located at the inner side of the solenoid structure (*Chook and Blobel, 2001*; *Cingolani et al., 1999*). Considering that the size of huntingtin is much bigger than other HEAT repeat proteins, we can imagine that the HEAT repeat domains can be folded back to form a closed structure that we have observed in huntingtin, having functional sites located in the internal cavity. This shape classifies huntingtin as a closed helical solenoid, contrasting with the shorter open curvilinear HEAT/HEAT-like repeat solenoids whose native structures have been solved at high resolution (*Cingolani et al., 1999*; *Groves et al., 1999*). Huntingtin's distinctive shape is predicted to provide both internal and external surface topologies that may mediate the binding of proteins or nucleic acids, as befitting a mechanical HEAT/HEAT-like repeat interaction-scaffold (*Takano and Gusella, 2002*).

Our biophysical analyses also provide basic insights for future higher-resolution studies that will be needed to more precisely delineate huntingtin structure and its modulation by the polyglutamine tract. The DSS-XL-MS intramolecular cross-linking data, together with our previously reported limited proteolysis analysis (*Seong et al., 2010*) provide a general sense of how rod-like α-helical HEAT/HEAT-like repeat domains, may fold to delimit the closed shape that we observe. The pattern of proteolysis, regardless of polyglutamine tract length, revealed a single major cleavage-sensitive site at ~ residue 1200, strongly predicting a major hinge or pivot that roughly parses the protein into two nearly equal 'arms', a 150 kDa NTD and a 200 kDa CTD. This is supported by the patterns of short- and mid-range intramolecular DSS-XL-MS delineated intramolecular contacts that are shared

by the Q23-, Q46- and Q78-huntingtin, which are mainly located within and between each of the regions that immediately flank this location, implying extensive local internal folding and close proximity of the 'arms' around the pivot-point. Multiple short- and mid-range interactions are also detected at the ends of the NTD-I and CTD-II. These contacts imply internal folding near the ends of each arm. Limited proteolysis of the amino-terminal domain did reveal an internal cleavage site, located at ∼ residue 500, which is consistent with internal pivot points, along with other internal folding, that may explain the short- and mid-range contacts detected by XL-MS near the terminus. However, the carboxyl-terminal domain lacked internal cleavage-accessible sites (*Seong et al., 2010*), implying a paucity of accessible hinge-points. Consistent with this, XL-MS fails to detect short- or mid-range cross-linked lysine residues in the sub-region spanning amino acids ∼1800 to ∼2300, except one long distance contact and one short distance contact only in Q46-huntingtin. This sub-region contains 13 lysine residues. It is possible that these residues are not DSS-accessible although we did observe DSS modified, but not cross-linked peptides in this region (data not shown), implying instead a more extended 3D structure that maximizes the surface area available for interaction with binding-partners, as in other HEAT repeat solenoid proteins (*Cingolani et al., 1999*; *Grinthal et al., 2010*; *Groves et al., 1999*). By contrast, the adjacent extreme C-terminus does display some internal short- and mid-range cross-linking interactions that suggest internal folding, though apparently without a well-defined proteolysis-accessible hinge-point. However, perhaps the most striking finding is the multiple long-range interaction cross-links between the CTD-II and CTD-I or NTD-II, as well as the long-range contacts of the NTD-I with the CTD-I close to the major hinge-pivot, which places the ends of each arm in close proximity to each other on the carboxyl-terminal domain arm, above the major hinge. Of note, EM analysis of the amino-terminal FLAG-antibody-huntingtin complex strongly suggests accessibility of the extreme N-terminus, and likely the adjacent polyglutamine tract, at the external surface. Thus, folding of the two main HEAT/HEAT-like domains forms an extensive internal cavity consistent with the shape that we observe in EM analysis, while providing an elegant explanation for the conundrum of how the polyglutamine tract located at the end of the amino-terminal arm may affect change throughout the entire protein.

All of the huntingtins, regardless of polyglutamine tract length, appear to have the same basic core structure. Our CD data demonstrates that they are all α-helical and have the same pattern of thermal stability, denaturing over the same temperature range. They have similar shapes in EM analysis and display mostly shared DSS-XL-MS intramolecular interactions, as discussed above. Nevertheless the differences among huntingtins with different polyglutamine lengths (Q23, Q46 and Q78) are telling. Each displays a unique intramolecular interaction pattern that is most evident in the long-range contacts detected by DSS-XL-MS analysis. Q23-huntingtin features unique additional contacts between the end of the amino-terminal domain arm with the carboxyl-terminal arm, near the hinge region, whereas Q78-huntingtin is characterized by additional interactions of the end region of the carboxyl-terminal arm with itself near the hinge region, in proximity to the location of the contacts made by the end region of the amino-terminal arm, or with the amino-terminal arm near the hinge. It is intriguing that most of these unique normal and mutant huntingtin folding characteristics involve contacts at the locations on the CTD-I near the major hinge where many other contacts that are common to both proteins converge. While Q23- and Q78-huntingtin show unique crosslinks skewed toward either amino-terminal or carboxyl-terminal region respectively, the wide-spread intramolecular interactions of Q46-huntingtin prompt us to hypothesize that the polyglutamine tract expansion induces subtle but progressive structural changes in huntingtin. This implies that polyglutamine tract length may subtly alter the basic structure by influencing the degree to which the end of the carboxyl-terminal arm is folded back upon itself onto the region around the major hinge: the longer the polyglutamine tract, the more are the contacts. This suggests a location of torsion or tension on the major hinge region that may be exerted by a balance between the positions of the amino-terminal end region contacts and carboxyl-terminal end region contacts. Consistent with our data, particularly including long-range interaction affected by the polyglutamine expansion, a few studies previously implied and reported the global structural and functional influence by the polyglutamine tract size: the interaction with HAP40 at the extreme carboxyl-terminus of huntingtin was influenced by the polyglutamine expansion (*Pal et al., 2006*); the proteolytically cleaved amino-terminal fragment interacted with the carboxyl-terminal fragment of huntingtin (*El-Daher et al., 2015*); the proximity between the first 17 residues and the polyproline region has been shown to change by the polyglutamine tract size both in exon 1 fragment and endogenous full-length huntingtin (*Caron et al.,*

*2013*). Perhaps huntingtin binding involves internal surface features that are accessed by a spring-loaded action of the major hinge-region that entails the end-region contacts of each arm. Importin HEAT/HEAT-like solenoid proteins undergo substantial conformational change around a hinge-pivot region upon cargo binding (*Cingolani et al., 1999*). The impact of the polyglutamine tract on huntingtin basic structural features strongly suggests that it is possible that huntingtin may also undergo dramatic conformational change upon interaction with its binding partners.

The success of our structural analysis using an allelic series of huntingtins with different polyglutamine tract lengths suggests that this approach applied to high-resolution analyses will continue to yield insights into the huntingtin disease trigger-mechanism.

## Materials and methods

### Human FLAG-huntingtin insect vector expression clones

All recombinant human FLAG-huntingtin cDNA used in this study were cloned in insect expression vector systems that were modified as previously described (*Seong et al., 2010*). Essentially, the original polylinker region of pFASTBAC1 vector (Invitrogen, Carlsbad, CA) was swapped with the modified polylinker containing 1X FLAG, 6X histidine tag, TEV protease recognition site, and several restriction enzyme sites, including NcoI, XhoI and SacII, using BamHI-KpnI sites. Full-length *HTT* cDNA was cloned in two steps. First, the NcoI-XhoI *HTT* cDNA fragment (*Faber et al., 1998*; *Seong et al., 2010*), encoding huntingtin amino acids 1–171 with varying polyglutamine tracts (Q2, 23, 46, 67, 78), was inserted between the unique NcoI and XhoI restriction sites in the modified linker. Second, the 9,046 bp XhoI-SacII *HTT* cDNA fragment from a full huntingtin cDNA clone, pBS-HD1-3144Q23 (*Faber et al., 1998*; *Seong et al., 2010*), encoding huntingtin amino acids 172–3,144 was inserted in frame using XhoI-SacII into the linker region. We confirmed by sequencing that this XhoI-SacII cDNA differs from the reference cDNA (Genbank accession number L12392) in two locations, reflecting polymorphisms: Lys1240Arg and the Delta2642 polymorphism (*Ambrose et al., 1994*) encoding Glu amino acids 2640–2645 in a run of five rather than six residues. The SacII site in the linker was unique because the original SacII site in pFASTBac1 vector had been removed before adding the linker. All final clones, namely pALHDQ2, pALHDQ23, pALHDQ46, pALHDQ67 and pALHDQ78 encoding full-length human FLAG-Q2-, 23-, 46-, 67- and Q78- huntingtin, respectively, were verified using full DNA sequence analysis. By convention, the amino acid numbering throughout the text follows the numbering of L12392 (Q23) regardless of the length of the polyglutamine tract.

### Full-length human huntingtin purification

Purification of FLAG-tag huntingtin was carried out as previously described (*Seong et al., 2010*). Briefly, FLAG-tag huntingtin was expressed from pALHD(Q2,23,46,67,78) in the Baculovirus Expression system (Invitrogen, Carlsbad, CA). The Sf9 cell lysate, obtained by freezing/thawing in buffer A (50 mM Tris-HCl pH 8.0, 500 mM NaCl, and 5% glycerol) containing complete protease inhibitor cocktail and PhosSTOP phosphatase inhibitor cocktail (Roche Applied Science, Branford, CT), was spun at 25,000 xg (2 hr). The supernatant was incubated with M2 anti-FLAG beads (Sigma-Aldrich, St. Louis, MO) (2 hr, 4°C). The non-specifically bound proteins were removed by washing extensively with buffer A. FLAG-huntingtin was eluted with buffer (50 mM Tris-HCl pH 8.0, 300 mM NaCl, 5% glycerol) containing 0.4 mg/ml FLAG peptide and loaded onto a calibrated Superose 6 10/300 column (GE Healthcare, Little Chalfont, UK) equilibrated with 50 mM Tris-HCl pH 8.0 and 150 mM NaCl. FLAG-huntingtin eluted discretely and was estimated to be at least 90% pure by Coomassie staining. To generate non-FLAG-tagged huntingtin, M2-bead bound huntingtin proteins were resuspended in buffer (20 mM HEPES, 150 mM NaCl, 0.5 mM EDTA, 0.25 mM DTT) and incubated with AcTEV protease (Invitrogen) for 5 hr at 25°C. The huntingtin proteins without FLAG-tag were released from the M2-bead and further purified using the same procedure as mentioned above.

Comparative analyses of huntingtin proteins with varying polyglutamine sizes were performed with an equal amount of each protein, verified by Bio-Rad DC protein assay (Bio-Rad Laboratories Inc, Hercules, CA) and R-250 Coomassie blue staining of bands on 10% SDS PAGE to control for potential differences in protein purity and amount. The molarity for all huntingtins was calculated using a molecular weight of 350 kDa deduced from the human cDNA sequence.

## Immunoblot analysis

50–100 ng of purified protein was run on a 10% Bis-Tris gel and transferred onto nitrocellulose membranes. All antibodies were blocked with 5% milk/TBST. Anti-huntingtin antibodies were used at dilutions of 1:2000 (mAB2166) and 1:5000 (HF-1). mAb1F8 antibody targeting the polyglutamine region was used at 1:10,000 dilution. After washing, the blots were probed with anti-Rabbit HRP secondary antibodies and developed using the ECL system. mAb2166 was purchased from Millipore (EMD Millipore, Darmstadt, Germany), whereas rabbit polyclonal antibodies HF-1 (against amino acids 1981–2580) were generated in the laboratory against the fusion protein, as previously reported (*Persichetti et al., 1995*; *Persichetti et al., 1996*). mAb1F8 antibody was also generated in the laboratory as previously reported (*Persichetti et al., 1999*). Streptavidin-HRP was obtained from Cell Signaling Technology (Danvers, MA).

## Circular dichroism

Purified full-length human huntingtins with different polyglutamine tract lengths (0.2 mg/ml) were dialyzed against 100 mM phosphate buffer pH 7.2 before CD analysis. Far-UV CD spectra were obtained by scanning from 260 nm to 185 nm at 25°C on a 410 AVIV spectropolarimeter (Lakewood, NJ) using a 1 mm quartz cuvette (Hellma, Plainview, NY) placed in a thermally controlled cell holder. The machine was equipped with a Peltier junction thermal device and a Thermo Neslab M25 circulating bath. Spectra were obtained with a wavelength step of 1 nm, an averaging time of 3 s for each data point and 30 s equilibration time between points. The data were calculated and plotted with Graphpad Prism software v.4.01. Concentrations of proteins were checked by absorbance at 280 nm prior to the experiment. The CD data was normalized for concentration to allow for a comparative analysis, and presented in the units of deg. $cm^2$/dmole. The thermal dependence of the CD was carried out for each protein by heating in 5°C steps from 25 to 95°C, with the wavelength set at 222 nm. The deconvolution of the CD curves to estimate secondary structure is not presented, as there is currently no reference database of HEAT-repeat proteins with known X-ray structure to make an accurate evaluation.

## Electron microscopy

Full-length FLAG-tag huntingtins were applied to ultracentrifugation at 74,329 xg for 16 hr with a 5–20% sucrose gradient in presence of a 0–0.2% glutaraldehyde gradient. A fraction containing only the monomeric huntingtin was collected and the protein was then negatively-stained with 2% (w/v) uranyl acetate for 2 min on 400 mesh carbon grids. Images were collected at 50,000x magnification with a defocus value of 0.5–1.5 µm on a 4x4K CCD camera (Tietz Vieo and imaging Processing System) attached to a Jeol JEM2100F filed emission gun transmission electron microscope at 200 kV. Data were processed using EMAN2 program (*Thakur et al., 2009*). For the huntingtin-FLAG Antibody complexes, FLAG-antibody (Sigma-Aldrich, St. Louis, MO) and huntingtin were incubated overnight at 4°C and only the antibody bound monomer of huntingtin was isolated as for huntingtin alone. Total 10,169, 9368, 4714, and 3239 particles were selected for Q23-huntingtin, Q78-huntingtin, Q23-huntingtin FLAG-antibody, and Q78-huntingtin FLAG-antibody, respectively. The selected particles were used for further processing to generate reference-free class-averages. The models were further iteratively refined with a low-pass-filter (cutoff=0.033). Four refined models were superimposed and difference maps between Q23-huntingtin, Q78-huntingtin alone and Q23-huntingtin-, Q78-huntingtin-Flag antibody complex, were calculated by Chimera (*Pettersen et al., 2004*), respectively.

## Cross-linking mass spectrometry analysis

In order to prepare the cross-linked huntingtins, Q23-, Q46- or Q78-huntingtin (200 µg, 1.0 mg/ml) were incubated with 1 mM of DSS-H12/D12 (Creative Molecules Inc.) for 20 min at 37°C with mild shaking. The cross-linking reaction was stopped by adding ammonium bicarbonate to a final concentration of 50 mM. Then, each cross-linked huntingtin was separated by ultracentrifugation at 111,541 xg for 16 hr with 10–30% sucrose gradient in 20 mM HEPES and 100 mM NaCl buffer. Only the monomeric population was collected (*Figure 4—figure supplement 1*) and evaporated to dryness for XL-MS analysis.

Approximately 50 µg of cross-linked huntingtins (Q23, Q46 and Q78) forms were separately redissolved in 75 µl 8 M urea. Potential disulfide bonds were reduced by addition of 5 µl of 50 mM tris(2-carboxyethyl)phosphine, followed by incubation for 30 min at 37°C, and free thiol groups were subsequently alkylated by the addition of 5 µl of a 100 mM iodoacetamide solution and incubation for 30 min at 23°C in the dark. For the two-step protease digestion, the samples were first diluted with 50 µl of 150 mM ammonium bicarbonate solution and 0.6 µg of endoproteinase Lys-C (Wako, Richmond, VA) was added. Lys-C digestion was carried out for 3 hr at 37°C. The samples were then further diluted by addition of 640 µl of 50 mM ammonium bicarbonate solution (final urea concentration = 1 M) and 1.2 µg of sequencing-grade trypsin (Promega, Madison, WI) was added. Trypsin digestion proceeded overnight at 37°C.

Enzymatic digestion was stopped by addition of pure formic acid to 2%, v/v, and samples were purified by solid-phase extraction (SPE) using 50 mg Sep-Pak tC18 cartridges (Waters, Milford, MA) using standard procedures. The SPE eluates were evaporated to dryness in a vacuum centrifuge. Digests of cross-linked huntingtins were fractionated by size exclusion chromatography (SEC) as described (*Leitner et al., 2012*; *2014*). Three fractions were collected and subjected to LC-MS/MS analysis on a Thermo Orbitrap Elite mass spectrometer as described previously (*Greber et al., 2014*). Cross-linked peptides were identified from the MS/MS spectra using xQuest (*Walzthoeni et al., 2012*) with the following settings: Enzyme = trypsin, maximum number of missed cleavages = 2, cross-linking site = K and mass shifts for the cross-linking reagent DSS-$d_0$/$d_{12}$. A sequence database was constructed from an independent search of an unfractionated sample against the UniProt/SwissProt database with Mascot (*Perkins et al., 1999*). The final database contained the huntingtin sequence and 13 identified low-level contaminants. xQuest search results were filtered according to the following criteria: mass error < 4 ppm, minimum peptide length = 6 residues, delta score < 0.9,% TIC $\geq$ 0.1, minimum number of bond cleavages per peptides = 4. An xQuest score cut-off of 17 was selected, corresponding to an estimated false discovery rate of < 5%. In addition to the cross-links on huntingtin, only one cross-link on a contaminant protein (HSP7C_DROME) was identified.

## Acknowledgements

We thank the members of the MacDonald, Lee, Song and Seong laboratories and Drs. S Kwak, R Lee, and D Lavery for suggestions and discussions. We are grateful to Jayalakshmi Mysore and Tammy Gillis (MGH NextGen Sequencing Core) for excellent technical support. This work was supported by CHDI Foundation Inc (JML and ISS); and National Institutes of Health/National Institute of Neurological Disorders and Stroke [R01 NS079651 to ISS]; Natural Sciences and Engineering Research Council of Canada [Grant 9848 to RE]; the European Research Council [Proteomics v. 3.0, ERC Advanced Grant 233226 to RA]; National Research Foundation of Korea [NRF-2013R1A1A2055605, NRF-2014K2A3A1000137, 2011-0020334, 2011-0031955 to JS]; and KIB [CMCC, N10150028 to JS]. ISS and JS were supported by Brain Pool program (152S-4-3-1328) by the Korean Federation of Science and Technology Society (KOFST) funded by the Korea Ministry of Science, ICT and Future Planning. T-YJ is supported by the Jonasson donation.

## Additional information

### Funding

| Funder | Grant reference number | Author |
|---|---|---|
| Natural Science and Engineering Research Council of Canada | Grant 9848 | Raquel Epand |
| Jonasson donation | | Tae-Yang Jung |
| CHDI Foundation | | Jong-Min Lee<br>Ihn Sik Seong |
| European Research Council | Proteomics v.3.0, ERC Advanced grant 233226 | Ruedi Aebersold |

| National Research Foundation of Korea | 2011-0020334 | Ji-Joon Song |
|---|---|---|
| National Research Foundation of Korea | NRF-2013R1A1A2055605 | Ji-Joon Song |
| KIB | CMCC, N10150028 | Ji-Joon Song |
| Korean Federation of Science and Technology Societies | Brain Pool program, 152S-4-3-1328 | Ji-Joon Song<br>Ihn Sik Seong |
| National Research Foundation of Korea | NRF-2014K2A23A 1000137 | Ji-Joon Song |
| National Research Foundation of Korea | 2011-0031955 | Ji-Joon Song |
| National Institute of Neurological Disorders and Stroke | R01 NS079651 | Ihn Sik Seong |

The funders had no role in study design, data collection and interpretation, or the decision to submit the work for publication.

## Author contributions

RV, J-JS, ISS, Conception and design, Acquisition of data, Analysis and interpretation of data, Drafting or revising the article, Contributed unpublished essential data or reagents; RE, ALe, BS, Acquisition of data, Analysis and interpretation of data, Drafting or revising the article, Contributed unpublished essential data or reagents; T-YJ, RJ, Acquisition of data, Analysis and interpretation of data, Drafting or revising the article; ALl, Acquisition of data, Contributed unpublished essential data or reagents; RSA, Acquisition of data, Analysis and interpretation of data; HL, J-ML, RA, HH, Analysis and interpretation of data, Drafting or revising the article

## Author ORCIDs

Ihn Sik Seong, http://orcid.org/0000-0003-4246-3356

## Additional files

### Supplementary files

• Supplementary file 1. The list of DSS modified peptides of human Q23- (A), Q46- (B) and Q78- (C) huntingtin identified in XL-MS. Huntingtin amino acid sequences of crosslinked peptides indicating the relative position of the linked lysines; Position1, Position2, absolute amino acid position of linked lysines.

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
