## [Decision Letter]

[Editors’ note: this article was originally rejected after discussions between the reviewers, but the authors were invited to resubmit after an appeal against the decision.]

Thank you for submitting your interesting work entitled "Huntingtin's spherical solenoid structure enables polyglutamine tract-dependent modulation of its structure and function" for consideration by *eLife*. Your article has been reviewed by three peer reviewers, and the evaluation has been overseen by a Reviewing Editor and a Senior Editor. Our decision has been reached after consultation between the reviewers. Based on these discussions and the individual reviews below, we regret to inform you that your work will not be considered further for publication in *eLife*.

As you can see the three referees found your work interesting but also raised a lot of questions that will require considerable work to be addressed. We anticipate that this will take too much time beyond the two months we allow for revisions and therefore we reject the paper in its current version. I hope that the reports are helpful and will give you guidance to improve your work. If you feel that you can address all questions with additional experimentation, you are allowed to resubmit the work to *eLife* but the paper will be considered as a new submission and reviewed as such.

*Reviewer #1:*

Protein structure is an important area of polyQ proteins/diseases where data are lacking. This study uses several approaches to examine structural features of recombinant human huntingtin. The most novel and informative studies are those using 3D EM and cross-linking followed by mass spec. While relatively crude these date provide the most informative structural data to date. Most of the work is done using huntingtin with either 23 or 78 glutamines – in a sense a range representing a broad range of allele lengths from wt to a rather long expanded allele, one that would cause juvenile HD. The extensive internal cavity is intriguing regarding potential functions of this protein. What is telling is the high degree of similarity on structure found for huntingtins with these two polyQ tract lengths. Yet subtle structural differences were seen that the authors speculate on in regards to polyQ effect on function and pathogenesis. What the authors fail to acknowledge is that the subtle changes were seen in comparing 23Q with 78Q. Much less data are provided for an intermediate allele of 46Qs – an allele representing a considerably more frequent mutant allele that leads to disease. Overall it seems that the structural effect of polyQ expansion is quite subtle, which in itself is an important finding. It would be interesting for the authors to comment on what the HTT work might imply for the other polyQ disorders.

Lastly, the fact that the NTD-1 region is in contact with essentially all of the others regions of HTT strongly indicates, as the authors mention, expansion of the polyQ in this region has the potential of impacting the rest of HTT. One suggestion would be to move this point from Figure 7 to Figure 4A so that Figure 7 would focus on interactions impacted by polyQ expansion.

*Reviewer #2:*

I appreciate the efforts in this study to generate three-dimensional (3D) models of full-length huntingtin (HTT) proteins with different Q lengths. Additionally, 16 novel phospho-specific huntingtin antibodies were produced in order to detect potential changes in the phosphorylation patterns in polyQ-containing huntingtin proteins. The study provides important information about potential intramolecular interactions in the full-length HTT protein. However, it is hard to judge the quality of the generated 3D models and the potential physiological relevance of the identified intramolecular interactions. A confirmation of the results with alternative methods is missing. Furthermore, there are several additional points that have to be addressed before the paper is suitable for publication in *eLife*.

1) The authors describe that the recombinant human huntingtin proteins with different polyglutamine lengths exhibit a very similar thermal stability (Results, first paragraph, Figure 1C). Judging from the results presented in Figure 1C I would conclude that Q46-huntingtin is less thermally stable than the Q23- and the Q78-huntingtin proteins. The Q46-huntingtin protein loses its secondary structure already at 35 °C whereas the Q23- and Q78-huntingtin proteins seem to be stable up to 45 °C. This needs to be clarified.

2) In the attempt to locate the N-terminus in Q23- and Q78 huntingtin the authors applied negative stain electron microscopy (subsection “3D EM analysis reveals a spherical shape with a central cavity and overlying N-terminus”, last paragraph, Figure 3). By comparing the huntingtin-FLAG-antibody complexes with the huntingtin protein alone I find it very hard to identify an extra density, which should indicate the FLAG-antibody bound to the N-terminus of huntingtin. I would highly appreciate additional experiments to support these results.

3) Further, the FLAG-tag might influence the location of the extreme N-terminus. As a TEV-cleavage site has been introduced the authors should remove the FLAG-tag from the HTT protein and apply a specific anti-HTT antibody (e.g. an antibody recognizing N17) to detect the N-terminus of huntingtin.

4) In Figure 4A the authors showed a hydrophobicity plot in order to support their choice of domain subdivision according to short rage intramolecular contacts. In my opinion this plot does not obviously support their choice. A different domain pattern might be conceivable judging from the hydrophobicity plot (for instance subdivision at aa 1201 and aa 2050).

5) The authors examined the differences of phosphorylation patterns in HTT with different polyglutamine lengths (subsection “The pattern of phosphorylated residues is altered with polyglutamine tract size“, Figure 5). It should be mentioned in the main text (not only in the Methods section) that these proteins are expressed and purified from Sf9 insect cells. Furthermore, I would like the authors to comment on the biological relevance of the identified phosphorylation patterns, as the proteins have not been purified from a mammalian system and only 14 out of 70 previously reported phosphorylation sites were confirmed/found.

6) In Figure 5—figure supplement 1A fifteen phosphopeptides are presented that were used for antibody generation. Although I highly appreciate the effort of generating and testing 16 different phospho-specific antibodies, I have concerns about phosphopeptide 11 and the consequent antibodies (α-Htt-p2114 and α-Htt-p2116). I would like the authors to explain, how they were able to purify two antibodies binding to different phospho-epitopes by the use of only one peptide comprising both of these epitopes.

7) Furthermore, in Figure 5—figure supplement 1B, 16 phosphopeptides are used for antibody testing. However, the numbers do not match the labeling in Figure 5—figure supplement 1A. Please clarify.

8) When analyzing Q-length dependent differences of phosphorylation (Figure 5—figure supplements 2 and 3) it is critical to assess whether the changes are significant. For better comparability, I suggest to display the quantification of all data sets using the same scale on the ordinate.

9) In the first paragraph of the subsection “Phosphorylation status distinguishes a novel property of mutant huntingtin “the authors stated that they analyzed pairs of phosphorylated and hypophosphorylated recombinant Q2-, Q23-, Q46- and Q78 huntingtin proteins by immunoblotting with phospho-epitope specific antibodies. Data are not shown for Q2- and Q46 huntingtin. Please clarify or show the missing data.

10) In Figure 6B, the quantification of H3 methylation does not seem to mirror the changes displayed in the autoradiogram above. This is especially true for measurements that have been done after CIP treatment. In comparison to their first publication in which the assay was introduced (Seong et al. 2010) the changes of H3 methylation, displayed in Figure 6B, are scarcely recognizable.

11) The authors stated that the phosphorylation status does not affect the secondary structure of either normal or mutant huntingtin (Figure 6—figure supplement 2). I would like to ask the authors to plot the MRE at 222 nm (in order to be able to compare the results to Figure 1) or comment why normalization is necessary for this data set.

*Reviewer #3:*

This manuscript presents data on the structure of full-length huntingtin and the results of polyglutamine expansion seen in Huntington's disease on that structure, and the effects on post-translational modifications of huntingtin. This work could be highly relevant and innovative to the HD community and any research focus on large proteins with repeated HEAT structures. This information fills a large gap in HD research due to the technical difficulties of studying a protein of 350Kda, but highlights the important observations of polyglutamine effects on huntingtin far beyond small fragments that have been used in mouse models of HD. The effects on post-translational modifications have direct relevance to HD therapeutic development.

The manuscript has significant problems in the writing, poor context to published work in this field, and some significant concerns about the biochemical systems used and the interpretations to this reviewer.

The paper needs a major writing revision, in the Abstract and title to reflect the mechanism outlined in the data. As it stands, the title and Abstract do not reflect the contents of the paper, i.e. the PRC complex data. The manuscript is poorly referenced on data concepts that have been reported in the past by others, but are being presented here as novel.

Structures for determination of the amino -terminal location used a FLAG tag. This peptide is commonly used for purification and immuno-tagging, but it is fused here to a short α-helical leader region before the polyglutamine tract and FLAG peptide is DYDDDDK, which now confers a huge charge on a region with a neutral charge. Thus, there is a good chance of artifactual effects from electrostatic interactions. FLAG tags are manipulated in biochemistry experiments to enhance the solubility of proteins, but they obviously cannot be innocuous with that run of charged residues to protein structure. It's not clear if the additionally charged polyhistidine tag and TEV protease site remain on the proteins or not.

From that work, "These observations strongly imply that the extreme N-terminus, and by inference the adjacent polyglutamine tract, is folded back,"

This concept was previously reported in PNAS in 2013 (Proc Natl Acad Sci U S A. 2013 Sep 3;110(36):14610-5.) but was not referenced in this manuscript. That manuscript also discusses the conformational change in the amino-terminus of huntingtin impacting total huntingtin conformation, which are presented here as novel concepts.

Similarly, the concept of the amino and carboxyl termini interacting is not novel, and has in fact been shown in vivo, with implications of huntingtin function at ER integrity. Again, not referenced.

(EMBO J. 2015 Sep 2;34(17):2255-71. doi: 10.15252/embj.201490808. Epub 2015 Jul 12.)

The data in Figure 3 is very difficult to interpret. They know polyglutamine expanded huntingtin has a tendency to precipitate, but while the data is the result of averaging of many images by EM, how to we know this isn't just precipitate versus soluble protein? I cannot distinguish the extra density of the FLAG-tag that the arrows are pointing to.

I'm really confused by the data in Figure 5. This is human recombinant huntingtin purified from insect Sf9 cells. How are any of the modifications in this figure therefore relevant in a mammalian context? For this to be true, then all of the modification in insects would have to be identical, despite over-expression of this protein, while we know stoichiometry of huntingtin is important. This is a problem with the manuscript, and the concept of the amino-terminus being important to total structure via PTMs, as the amino-terminus of mammalian huntingtin proteins has no homology to the gene annotated in insect species as huntingtin. For all we know, those PTMs may be relevant for proper folding, then removed, but the purification is in the context of phosphatase inhibitor cocktails. This leaves me with significant concerns about the validity of a data in Figure 5 to mammalian context.

While the PRC2 complex was assembled in equimolar concentration, I am surprised that this complex in vitro can be considered relevant to biology when neither DNA nor chromatin is present. I don't think the exact minimalist nature of this experiment has been outlined with the inherent caveats. They are describing a polycomb repressor complex that acts on chromatin and DNA, in the absence of chromatin or DNA. To stay within this manuscript, they will need cell data.

The Discussion needs revision. The term "function" is not appropriate in the second paragraph. They show a disrupted interaction with EZH2, but at no point in this manuscript is actual function described, and they cannot conclude functional information for reasons outlined above.

They examined 16 phospho-sites across huntingtin, but have not tested the most studied site with the first 17 amino acids, which has been shown by genetics and small molecule effects to affect the disease phenotype in mouse models (and by genetic modification, to thus be the most critical site). They have not tested every PTM in regions of huntingtin that are known as fragments to cause phenotype in the mouse. This is a major omission in this data. They did reference this work by Gu et al. The problem is that they have no data in a region of huntingtin that is known to cause a disease phenotype in trans in a polyglutamine -length dependent manner.

How many of those phospho-antibodies have been fully validated? The data is not shown, and no figure supplements 1,2 and 3 were uploaded. I can only access the table. The full gels should be supplemental data, not just the cropped images.

"Thus folding of the two main HEAT/HEAT-like domains forms an extensive internal cavity consistent with the shape that we observe in EM analysis, while providing an elegant explanation for the conundrum of how the polyglutamine tract located at the end of the N-terminal arm may affect change throughout the entire protein.": I fail to see this elegant explanation from Figure 7. They need a clearer model. Most HEAT importins show a super-helical structure with the internal face interacting with proteins to induce allosteric effects on the HEAT protein to modify the scaffold that are transduced along the scaffold (as described by Kleckner in one of the references). This has been done very well by Yuh-min Chook on the analysis of karyopherin Beta2, a huntingtin interactor and HEAT-rich protein.

*Reviewer #3 (Additional data files and statistical comments):*

Need to see phospho-antibody validation data. This would include: dot blots to gauge affinity, full western blots to gauge specificity, as well as blots on extracts with either no or reduced huntingtin, and IF studies with antigen peptide competition.

[Editors’ note: what now follows is the decision letter after the authors submitted for further consideration.]

Thank you for resubmitting your work entitled "Huntingtin's spherical solenoid structure enables polyglutamine tract-dependent modulation of its structure and function" for further consideration at *eLife*. Your revised article has been favorably evaluated by a Senior editor, a Reviewing editor, and three reviewers. The study presents the results of structural studies on full-length huntingtin using CD, 3D EM, and a mass spec/cross-linking approach. In this revised manuscript the investigators include additional data on huntingtin with 46 repeats, a repeat tract found frequently on affected alleles. Finding that huntingtin folds into a structure that consists of a large central cavity, regardless of polyQ tract length, is intriguing regarding potential functions of this large protein and effect of polyQ expansion on function. Overall this work is an important contribution to HD research and polyQ diseases in general, field where structural data are lacking. However, the manuscript still needs important revisions. We feel sorry that we have to delay further acceptance, but important advice given in the first round of revision is insufficiently addressed to allow publication without further important adaptations of the text.

The referees raised mainly doubts with regard to the phosphorylation experiments and in particular the physiological relevance of these data. There are about 100 kinases in fly, around 200 in worm, and over 500 in humans, with 13 human kinase families not seen in fly or worm (Manning et al., 2002. TIBS 27(10) pp.514-520). Thus, there is a high chance of different phospho-PTMs of a human protein expressed in insect cells. A useful reference of a study that showed only 38% similarity of PTMs from human to *Drosophila* cells: Krishnamoorthy, S. PLoS ONE. 2008; 3(8): e2877. Positively one could argue that even if these phosphorylation events are physiologically not validated, they provide proof of concept that changes in the N-terminal extension can affect posttranslational modification (PTM). This needs however follow up studies in a mammalian expression system for further validation. We suggest to the authors to consider to remove this part of the manuscript or to be at least much more careful in the claims. They should clearly mention that insect cells are not a full model for PTM in mammalian cells and that they only cover a fraction of the kinome in mammals. They also should explicitly mention the possibility of artefacts. Finally, we provide several references of papers that investigated the topic before, these papers need to be discussed in a final version. Alternatively, this part can be deleted as there remained also still serious doubts about the quality controls of the phosphoantibodies which were provided in Figure 5—figure supplement 5 but for which no controls on reactivity in cell extracts were provided.

There are, as indicated already, also additional serious problems with the representation of past research. We suggest to discuss and compare previously published data with the data/conclusions in the current paper using the papers and indications provided in additional comments.

Additional comments:

1) In Figure 1C the meaning of the dotted line is not explained. Please include one sentence for clarification as you did in the original version of the paper.

2) In the figure legend of Figure 1C exchange duplicated for duplicates.

3) From the Introduction: "Indeed, a comparison of lines of transgenic modified HTT BAC mice has implicated unique amino-terminal serine phosphorylation in mutant huntingtin gain of toxic function, indirectly implying a long-range impact of the amino-terminal region on huntingtin structure and function (Gu et al., 2009)." The Gu et al. manuscript clearly demonstrates the protective effect of serine 13 and 16 phospho-mimetic mutations in a Q84 context, and no effect of S13AS16A mutations. This is the opposite of this statement. Furthermore, another group showed protection in the YAC128 model by small molecules that induced this PTM. The proper reference for polyglutamine effects at distal regions of huntingtin is likely from Zerial's work on full-length huntingtin and HAP40 interactions at the carboxyl-terminus affected by the polyglutamine expansion. (Pal et al., J Cell Biol. 2006 Feb 13;172(4):605-18).

Another paper that deserves discussion in this contest is Schilling B, Gafni J, Torcassi C, Cong X, Row RH, LaFevre-Bernt MA, Cusack MP, Ratovitski T, Hirschhorn R, Ross CA, Gibson BW, Ellerby LM. J Biol Chem. 2006 Aug 18;281(33):23686-97. Epub 2006 Jun 16. This work mapped out phosphorylation sites across huntingtin, and showed a trend of all sites being hypo-phosphorylated due to polyglutamine expansion, even near the carboxyl-terminus.

4) The authors claim in the Abstract that they provide the first glimpse into the structural properties of huntingtin and an elegant solution to the apparent conundrum of how the extreme amino-terminal polyglutamine tract confers a novel property on huntingtin, causing the disease. Some qualifiers or reformulation are indicated: this is not the first glimpse, a recent huntingtin structural paper demonstrated the α-solenoid structure of huntingtin. There are across a few labs many papers on huntingtin PTMs in full-length huntingtin and conclusions of reduced phosphorylation at well characterized sites. Many publications suggest a loss of function of mutant huntingtin in events post-development. Some are included above and below for the reference of the authors:

Pal A, Severin F, Lommer B, Shevchenko A, Zerial M. J Cell Biol. 2006 Feb 13;172(4):605-18. Here they showed that interactions with HAP40 at the

extreme c-terminus was influenced by the polyglutamine expansion on the carboxyl-terminus.

El-Daher MT, Hangen E, Bruyère J, Poizat G, Al-Ramahi I, Pardo R, Bourg N, Souquere S, Mayet C, Pierron G, Lévêque-Fort S, Botas J, Humbert S, Saudou F. EMBO J. 2015 Sep 2;34(17):2255-71. doi: 10.15252/embj.201490808. Epub 2015 Jul 12. This work showed that Amino-terminal proteolytic fragments could interact with Carboxy-terminal fragments of huntingtin, as well as toxicity by the carboxyl terminal fragments of huntingtin.

Caron NS, Desmond CR, Xia J, Truant R. Proc Natl Acad Sci U S A. 2013 Sep 3;110(36):14610-5. doi: 10.1073/pnas.1301342110. Epub 2013 Jul 29. This work showed the folding back of huntingtin amino terminus to carboxyl-distal regions of huntingtin in both fragment and full-length endogenous huntingtin contexts.

5) One of the referees questioned the data in Figure 3. S/he suggested "that the images were selected with the nanogold particles on the top in B (which are not as specific as gold-labelled antibodies), but it is difficult to gauge this relative to the rest of the structures, which all look very different. In my opinion, the data is not consistent with the 2D class average in 3D structures, and could just be the selection of images of a different orientation with the epitope up in the Z plane.

In 3A, I see obvious differences in the 2D-3D class average with or without the antibody in the top half of the structure."

Please address this concern in the new revision of the paper.

6) Some discussion of the structural data comparing with other HEAT-rich proteins would be beneficial. Most of these HEAT proteins have large, solvent exposed central cavity, this is very common in the nuclear transport field. So, in a large HEAT protein, this would be anticipated.

7) Figure 5: Mab2166 is a good choice of antibody, as it is one of the few fully validated anti-human huntingtin antibodies. They need a loading control, or proper loading that has the Mab2166 signal within the linear range of the assays. This is a problem in 5C, the bands are too intense.

8) In Figure 6A, the first 2166 blots are also all darker than the range of the assay, and in the bottom blot comparing to pS2550, the levels of pS2550 clearly track with more or less total huntingtin in the 2166 blot (lanes 1 and 3). In both figures, they would be far more convincing with some additional cell biological imaging data in addition to the biochemical assays. This would give the reader more confidence.

---

## [Author Response]

[Editors’ note: the author responses to the first round of peer review follow.]

*Reviewer #1:*

[…]

*Much less data are provided for an intermediate allele of 46Qs – an allele representing a considerably more frequent mutant allele that leads to disease. Overall it seems that the structural effect of polyQ expansion is quite subtle, which in itself is an important finding. It would be interesting for the authors to comment on what the HTT work might imply for the other polyQ disorders.*

We fully agreed and performed XL-MS analysis on Q46-huntingtin. The pair-comparison of the XL-MS data from Q23-, Q46- and Q78-huntingtin show that while Q23-and Q78-huntingtin show unique crosslinks skewed at either N-terminal or C-terminal region respectively, Q46-huntingtin shows widespread intramolecular interaction throughout the entire region of huntingtin as if Q46-huntingtin posits an intermediate conformation between Q23- and Q78-huntingtin. These analyses suggest that the polyglutamine expansion located at the very N-terminal induces progressive but subtle structural change in full-length huntingtin in a polyglutamine length dependent manner. We incorporated the new data in a reformatted form of Figure 4, its figure supplement, Figure 4—figure supplement 1 and [Supplementary-material SD1-data], and describe the results in a revised Results, Discussion and Methods.

*Lastly, the fact that the NTD-1 region is in contact with essentially all of the others regions of HTT strongly indicates, as the authors mention, expansion of the polyQ in this region has the potential of impacting the rest of HTT. One suggestion would be to move this point from Figure 7 to Figure 4A so that Figure 7 would focus on interactions impacted by polyQ expansion.*

We agree and moved the cartoon in Figure 7 including a new Q46-huntingtin schematic to Figure 4E showing the differential intramolecular interactions as the polyglutamine tract expands. We removed Figure 7 because we can describe the point about the global impact of polyglutamine expansion on huntingtin structure and function in the revised Results and Discussion with Figure 4E.

*Reviewer #2:*

*1) The authors describe that the recombinant human huntingtin proteins with different polyglutamine lengths exhibit a very similar thermal stability (Results, first paragraph, Figure 1). Judging from the results presented in Figure 1 I would conclude that Q46-huntingtin is less thermally stable than the Q23- and the Q78-huntingtin proteins. The Q46-huntingtin protein loses its secondary structure already at 35 °C whereas the Q23- and Q78-huntingtin proteins seem to be stable up to 45 °C. This needs to be clarified.*

We agree and have clarified by explicitly stating that there are variations caused by inefficient mixing in the cuvette, inherent in taking readings every five degrees of heating, and stating that the heat denaturation curves with the huntingtin proteins were duplicated, providing confidence that their profiles are not significantly different, starting their denaturation above 40 ^o^C. We modified Figure 1 to enlarge the error bars by changing the scale of y-axis and revised the legend of Figure 1 in the revised manuscript.

*2) In the attempt to locate the N-terminus in Q23- and Q78 huntingtin the authors applied negative stain electron microscopy (subsection “3D EM analysis reveals a spherical shape with a central cavity and overlying N-terminus”, last paragraph, Figure 3). By comparing the huntingtin-FLAG-antibody complexes with the huntingtin protein alone I find it very hard to identify an extra density, which should indicate the FLAG-antibody bound to the N-terminus of huntingtin. I would highly appreciate additional experiments to support these results.*

*3) Further, the FLAG-tag might influence the location of the extreme N-terminus. As a TEV-cleavage site has been introduced the authors should remove the FLAG-tag from the HTT protein and apply a specific anti-HTT antibody (e.g. an antibody recognizing N17) to detect the N-terminus of huntingtin.*

We agree and have now performed several experiments to locate the N-terminal region of huntingtin. In the revised manuscript, we collected for huntingtin alone (no antibody) more micrograph images and reprocessed to reconstitute 3D EM structures of huntingtin alone which by comparison with the results for the huntingtin-antibody complexes clearly show the extra-density of the FLAG antibodies in the latter but similar shape for huntingtin alone and huntingtin-antibody complexes, indicating that the FLAG-antibody does not evidently alter full-length huntingtin EM structure. Thus, we have updated Figure 2 and added the 3D reconstituted images of huntingtin-antibody complexes with their 2D averages to compare with ones of huntingtin alone in the revised Figure 3.

The reviewer suggests using an ‘N17 antibody’ to locate the N-terminal region of full-length native huntingtin. We have attempted this experiment with the N17 antibody (from Dr. Ray Truant) and have repeatedly found that this reagent does not bind to native (undenatured) full-length huntingtin, either with or without the FLAG-tag, although it does bind well when full-length huntingtin proteins are denatured and analyzed by immunoblot or by immunocytochemistry. Dr, Truant had not tried to detect native huntingtin with this reagent (personal communication) but our results suggest either that the N17-epitope is buried in the native full-length huntingtin structure or, alternatively that the N17-antibody epitope is formed only when the protein is denatured and assumes a non-native structure. The latter explanation seems more likely, given the accessibility of the FLAG-epitope in native FLAG-full-length huntingtin and the fact that the N17 antibody was generated to denatured N-17 peptide. In any case, the inability of the N17 antibody to bind to native full-length huntingtin precluded our ability to utilize the N17 antibody to locate the N-terminal of full-length ‘native’ huntingtin. Instead, as the reviewer suggested, we applied an additional experimental method to locate the N-terminal of huntingtin. We utilized the Ni-NTA-Nanogold for labeling His-tag located at the N-termini of Q23- and Q78-full-length huntingtins, and observed that the gold nanoparticle labeled Q23- and Q78-huntingtins, permitting the N-termini to be visualized. These data are also included in the revised Figure 3. With these updated and new data, we have clearly described the location near the N-terminal of huntingtin in our EM image in the revised Result section.

*4) In Figure 4 the authors showed a hydrophobicity plot in order to support their choice of domain subdivision according to short rage intramolecular contacts. In my opinion this plot does not obviously support their choice. A different domain pattern might be conceivable judging from the hydrophobicity plot (for instance subdivision at aa 1201 and aa 2050).*

We agree. As stated in the text, the sub-domains are delineated based on the short-range contacts. We were attempting to make the point that at each sub-domain-edge there is a transition in hydrophobicity. We have removed the hydrophobicity figure from Figure 4 and simply mentioned the observation of the transition in hydrophobicity prediction at the edge of each subdomain (data not shown) in the revised manuscript.

*5) The authors examined the differences of phosphorylation patterns in HTT with different polyglutamine lengths (subsection “The pattern of phosphorylated residues is altered with polyglutamine tract size“, Figure 5). It should be mentioned in the main text (not only in the Methods section) that these proteins are expressed and purified from Sf9 insect cells. Furthermore, I would like the authors to comment on the biological relevance of the identified phosphorylation patterns, as the proteins have not been purified from a mammalian system and only 14 out of 70 previously reported phosphorylation sites were confirmed/found.*

This comment indicates that we did not properly contextualize the utility of the insect cell system, for which there is deep precedent for studying mammalian PTMs. We have therefore now emphasized the insect cell system as an excellent surrogate for mammalian cell biology in the revised Results section by explicitly explaining the conservation of mammalian kinase families in insect cells with a reference (Busconi, L and Michel,T (1995) “Recombinant endothelial nitric oxide synthase: post-translational modifications in a baculovirus expression system” Mol Pharmacol 47(4): 655-659). We have now also explicitly mentioned in the revised manuscript that 14 out of the 16 phosphorylation sites that we have identified are also found in mammalian cell studies. The number ‘70’ noted by the Reviewer is the sum total of sites reported in more than 10 different studies, each of which each reports 10 ~ 15 sites, not all of which are consistent from study-to-study. To clarify this point, we added “collectively” in the first paragraph of the subsection “The pattern of phosphorylated residues is altered with polyglutamine tract size“.

*6) In Figure 5—figure supplement 1A fifteen phosphopeptides are presented that were used for antibody generation. Although I highly appreciate the effort of generating and testing 16 different phospho-specific antibodies, I have concerns about phosphopeptide 11 and the consequent antibodies (α-Htt-p2114 and α-Htt-p2116). I would like the authors to explain, how they were able to purify two antibodies binding to different phospho-epitopes by the use of only one peptide comprising both of these epitopes.*

*7) Furthermore, in Figure 5—figure supplement 1B, 16 phosphopeptides are used for antibody testing. However, the numbers do not match the labeling in Figure 5—figure supplement 1A. Please clarify.*

We appear not to have written clearly, leading to confusion. Figure 5—figure supplement 1A reports a list of phosphopeptides that were detected by mass spectrometry, as stated in the figure legend, not the peptides used for immunization for production of phosphor-site specific antibodies. As stated in the Methods, 16 different phosphor-antibodies against 16 identified sites (not phosphopeptides) were generated to validate each of the 16 phosphorylation sites shown in Figure 5B. To clarify further, we have revised “~ the respective phosphopeptides” into “~ the 16 respective phosphopeptides having single phosphorylation site (e.g. either pS2114 or pS2116)” in Figure 5—figure supplement 1B legend.

*8) When analyzing Q-length dependent differences of phosphorylation (Figure 5—figure supplements 2 and 3) it is critical to assess whether the changes are significant. For better comparability, I suggest to display the quantification of all data sets using the same scale on the ordinate.*

We agree and have changed the scales for the quantification data in Figure 5 and Figure 5—figure supplement 3 so that the same scale is used in both.

*9) In the first paragraph of the subsection “Phosphorylation status distinguishes a novel property of mutant huntingtin “the authors stated that they analyzed pairs of phosphorylated and hypophosphorylated recombinant Q2-, Q23-, Q46- and Q78 huntingtin proteins by immunoblotting with phospho-epitope specific antibodies. Data are not shown for Q2- and Q46 huntingtin. Please clarify or show the missing data.*

We agree and to be accurate we have now replaced “The latter huntingtins” with “Two representative hypophosphorylated Q23- and Q78-huntingtin proteins” in the revised manuscript.

*10) In Figure 6B, the quantification of H3 methylation does not seem to mirror the changes displayed in the autoradiogram above. This is especially true for measurements that have been done after CIP treatment. In comparison to their first publication in which the assay was introduced (Seong* et al. *2010) the changes of H3 methylation, displayed in Figure 6B, are scarcely recognizable.*

The autoradiogram is one representative from 3 independent experiments that were quantified and summarize in the graph below the autoradiogram. We have included all three autoradiograms with each band quantification value including the other two as a supplementary figure (Figure 6—figure supplement 3).

*11) The authors stated that the phosphorylation status does not affect the secondary structure of either normal or mutant huntingtin (Figure 6—figure supplement 2). I would like to ask the authors to plot the MRE at 222 nm (in order to be able to compare the results to Figure 1) or comment why normalization is necessary for this data set.* We agree. Figure 6—figure supplement 2 cannot be properly compared with Figure 1 because we found that the additional steps, such as the phosphatase treatment, required to prepare hypophosphorylated huntingtin (described in Methods) exert effects on MRE at 222nm, as well as the sensitivity of CD measurement to batch effects. Thus, in Figure 6—figure supplement 2 legend in the revised manuscript we have added the comment, “To examine a subtle potential difference, each hypophosphorylated huntingtin was prepared together with its counterpart from the beginning to the end of purification including the additional phosphatase treatment step (described in Method) and the results were normalized by each counterpart’s MRE value.”

*Reviewer #3: The manuscript has significant problems in the writing, poor context to published work in this field, and some significant concerns about the biochemical systems used and the interpretations to this reviewer. The paper needs a major writing revision, in the Abstract and title to reflect the mechanism outlined in the data. As it stands, the title and Abstract do not reflect the contents of the paper, i.e. the PRC complex data. The manuscript is poorly referenced on data concepts that have been reported in the past by others, but are being presented here as novel.*

We are perplexed by these comments and other related comments (below), but speculate from the emphasis placed on the exon 1-fragment literature, which is not relevant to our study, that the Reviewer may have: 1) been misled by the ‘huntingtin literature’ which in reality reports only on ‘huntingtin-fragment which is inaccurately called ‘huntingtin’; 2) not fully understood our genetics-based biochemical strategy for full-length huntingtin, which is a classic structure-function approach; 3) thought that we were evaluating the standard fragment-hypothesis rather than the specific genetic hypothesis that we are evaluating (i.e. the fundamental HD mutational mechanism entails an impact of the polyQ tract at the N-terminus of full-length huntingtin that can be observed to modulate the structure and activity of the full-length huntingtin protein in a polyQ length-dependent manner), and finally 4) not appreciated that our biochemical PRC2 nucleosome array histone H3K27me3 activity assay has previously been validated as a readout for full-length huntingtin functional activities in murine development in vivoand in cell culture, as reported in two previous publications (Seong et al., Hum. Mol. Genet. 2010;19:573-583 and Biagioli et al., Hum Mol Genet. 2015; 24:2375-2389), as referenced in our manuscript. Our observations on full-length huntingtin are novel and we do not cite reports studying exon 1-fragment because the data has not been demonstrated to be predictive of the structure of the polyQ region in full-length huntingtin structure or of the normal functional activity of full-length huntingtin. Indeed, the fragment lacks 97% of all full-length huntingtin residues, has none of full-length huntingtin’s HEAT repeats, and does not have the normal functional activities of full-length huntingtin. Thus, we strongly rebut most comments of this reviewer but also respond to the comments which are feasible and strengthen our manuscript as much as we can (detailed below in each point).

*Structures for determination of the amino -terminal location used a FLAG tag. This peptide is commonly used for purification and immuno-tagging, but it is fused here to a short α-helical leader region before the polyglutamine tract and FLAG peptide is DYDDDDK, which now confers a huge charge on a region with a neutral charge. Thus, there is a good chance of artifactual effects from electrostatic interactions. FLAG tags are manipulated in biochemistry experiments to enhance the solubility of proteins, but they obviously cannot be innocuous with that run of charged residues to protein structure. It's not clear if the additionally charged polyhistidine tag and TEV protease site remain on the proteins or not.*

We agree that it is possible that the FLAG tag may introduce ‘artifact’ relative to full-length huntingtin without the tag. To address this (as discussed above Reviewer 2 comments), we have generated 2D class-averages of huntingtin without the tag and added in the revised Figure 2—figure supplement 3 which shows similar structural properties as FLAG-huntingtin, thereby demonstrating that the FLAG-tag does not introduce significant artifact.

*From that work, "These observations strongly imply that the extreme N-terminus, and by inference the adjacent polyglutamine tract, is folded back," This concept was previously reported in PNAS in 2013 (Proc Natl Acad Sci U S A. 2013 Sep 3;110(36):14610-5.) but was not referenced in this manuscript. That manuscript also discusses the conformational change in the amino-terminus of huntingtin impacting total huntingtin conformation, which are presented here as novel concepts.*

In fact, this study reports neither. In that study, FRET assay was designed to estimate the distance between the N17 region and the polyproline region, which flank the polyglutamine stretch, in huntingtin-exon 1-fragment. They first found that in the exon 1 fragment, the polyglutamine region folds so that the two flanking regions come into contact in a polyQ length dependent manner. They also showed that the FRET signal from this contact between the N17 region and the polyproline region in exon 1 is different in HD fibroblast compared to control fibroblast. Thus, this work did not show or discuss the possibility that the extreme N-terminus of full-length huntingtin is physically folded back to contact to the C-terminal-region of full-length huntingtin. Instead, what we mean by ‘folding back’ is based on our observations of the closed spherical shape of 3D EM images and the long-range interactions of the NTD-1 region of full-length huntingtin with a region of full-length huntingtin that is 1000 amino acids away in the primary amino acid sequence. Moreover, we did not present our model as a novel concept because folding back of HEAT-repeat domains is a concept already in the literature, but we do state that the model that we present for full-length huntingtin that is based on our empirical findings is novel, because it is the first report of a structure-based analysis of the full-length huntingtin protein. We have in the Discussion cited the PNAS paper as the local interaction changes by the polyQ tract size in fragment, while stating that this is quite distinct from our long-range interactions affected by the polyQ tract of the full-length protein.

*Similarly, the concept of the amino and carboxyl termini interacting is not novel, and has in fact been shown in vivo, with implications of huntingtin function at ER integrity. Again, not referenced. (EMBO J. 2015 Sep 2;34(17):2255-71. doi: 10.15252/embj.201490808. Epub 2015 Jul 12.)*

What this study actually showed was the folding structure of a large carboxyl terminal fragment (~2,500 amino acids), which after cleavage from full-length huntingtin is toxic. The interaction shown was between C-terminal fragment and a specific N-terminal fragment (~500 amino acids), but not a small N-terminal (~ 100 amino acids). The study did not show this interaction (or any other) in the context of full-length huntingtin, which would be required to make the observations relevant to our full-length huntingtin study. However, to clarify for the readers, we have in the revised Discussion cited this paper stating how as fragments which have different structural features than the regions as found in full-length huntingtin, can interact, but the interactions are not found in our full-length huntingtin data.

*The data in Figure 3 is very difficult to interpret. They know polyglutamine expanded huntingtin has a tendency to precipitate, but while the data is the result of averaging of many images by EM, how to we know this isn't just precipitate versus soluble protein? I cannot distinguish the extra density of the FLAG-tag that the arrows are pointing to.*

First, it is important to note that the full-length huntingtin protein does not have the same strong propensity to aggregate as N-terminal-fragment does. We clearly demonstrate that the EM analysis of full-length huntingtin is done after the separation of antibody bound monomer fraction by GraFix (legend Figure 3) and confirm this with micrographs in the Figure 2—figure supplement 3, which show disperse particles of the homogenous monomer population separated by GraFix. We fully agree with the second point and clarified the extra density of the FLAG-tag location with the new experiment using Ni-NTA-Nanogold in the revised Figure 3 (See response to Reviewer #2).

*I'm really confused by the data in Figure 5. This is human recombinant huntingtin purified from insect Sf9 cells. How are any of the modifications in this figure therefore relevant in a mammalian context? For this to be true, then all of the modification in insects would have to be identical, despite over-expression of this protein, while we know stoichiometry of huntingtin is important. This is a problem with the manuscript, and the concept of the amino-terminus being important to total structure via PTMs, as the amino-terminus of mammalian huntingtin proteins has no homology to the gene annotated in insect species as huntingtin.*

To address the concern of PTM sites identified in insect cells we have now added a reference to a study that demonstrates the fundamental conservation of the insect phosphorylation system with the mammalian system and emphasized in our text that the majority of full-length human huntingtin PTMs identified in our study have previously been reported from studies of full-length mammalian huntingtin (see response to Reviewer #2). Our study is of human full-length huntingtin, not of insect huntingtin, so the comment on lack of homology is not relevant to our study.

*For all we know, those PTMs may be relevant for proper folding, then removed, but the purification is in the context of phosphatase inhibitor cocktails. This leaves me with significant concerns about the validity of a data in Figure 5 to mammalian context.*

We are not sure that the Reviewer means with this comment. There are cases that PTM is necessary for proper folding of proteins. In our experiments, we treated after the protein was expressed in cell and purified. It is unlikely that removing PTM from already well-folded protein induces protein-misfolding. In addition, as we mentioned in our response to Reviewer #2 and above, the conservation of mammalian kinases in insect cells and the confirmation in mammalian cells of 14 sites out of 16 sites identified from our purified proteins clearly confirm that the phosphosites identified in full-length huntingtin expressed in insect cells are indeed fully relevant to (and indeed are the same as) the phosphosites found in a mammalian cell context.

*While the PRC2 complex was assembled in equimolar concentration, I am surprised that this complex in vitro can be considered relevant to biology when neither DNA nor chromatin is present. I don't think the exact minimalist nature of this experiment has been outlined with the inherent caveats. They are describing a polycomb repressor complex that acts on chromatin and DNA, in the absence of chromatin or DNA. To stay within this manuscript, they will need cell data.*

We clearly state in the manuscript: 1) that the PCR2 assay is comprised of DNA/chromatin in the form of nucleosomal arrays and 2) that this ‘minimalist’ biochemical assay has previously been validated as a read-out for full-length huntingtin normal function (and effect of its polyQ tract) on PRC2-dependent chromatin marks and regulation as demonstrated both in vivo (Seong et al. 2010) and in cell culture studies (Biagioli et al. 2015). We emphasized both points in the revised manuscript by revising the descriptions already in the manuscript “we then assessed the functional activities of the entire series of phosphorylated and hypophosphorylated huntingtin pairs in our PRC2-dependent nucleosome-array histone H3 lysine 27 trimethylation (histone H3K27me3) assay (Seong et al., 2010)” in Results, and the statements that the nucleosomal array consists of G5E4 DNA with 12 nucleosome in Methods.

*The Discussion needs revision. The term "function" is not appropriate in the second paragraph. They show a disrupted interaction with EZH2, but at no point in this manuscript is actual function described, and they cannot conclude functional information for reasons outlined above.*

Our PRC2-activity assay is a validated readout for normal full-length huntingtin function in PRC2-dependent regulation of chromatin histone H3K27me3 mark and chromatin regulation, as previously demonstrated. As above, we further emphasized that this full-length huntingtin function has been previously shown.

*They examined 16 phospho-sites across huntingtin, but have not tested the most studied site with the first 17 amino acids, which has been shown by genetics and small molecule effects to affect the disease phenotype in mouse models (and by genetic modification, to thus be the most critical site). They have not tested every PTM in regions of huntingtin that are known as fragments to cause phenotype in the mouse. This is a major omission in this data. They did reference this work by Gu et al. The problem is that they have no data in a region of huntingtin that is known to cause a disease phenotype in trans in a polyglutamine -length dependent manner.*

This hypothesis is very different to the genetic-based hypothesis that we are actually testing in our study (the polyglutamine tract in full-length huntingtin will be found to confer altered structural features on the full-length huntingtin protein and altered normal full-length huntingtin functional activity). It is not our purpose to look at toxic-fragments or to look at PTMs previously identified. It is our purpose to conduct a systematic evaluation of polyQ tract expansion on phosphorylation of full-length human huntingtin. We do cite Gu et al. 2009 (S13/16 phosphorylation) but in the revised Discussion clarified that in none of the three full-length huntingtin proteins analyzed did we find phosphorylation sites in the extreme N-terminal region of the full-length protein, a result that may reflect either differences between N-terminal-fragment and full-length huntingtin or may reflect the difficulty of identifying amino terminal PTM using mass spec or both.

*How many of those phospho-antibodies have been fully validated? The data is not shown, and no figure supplements 1,2 and 3 were uploaded. I can only access the table. The full gels should be supplemental data, not just the cropped images.*

The specificity of each antibody compared to each of the other 15 phospho-peptides including each corresponding (the same sequence) non-phospho peptide is clearly demonstrated in Figure 5—figure supplement 1B and the confirmation by phosphatase treatment in Figure 6—figure supplement 1. The full gels of western blot data with 16 phospho-antibodies in Figure 5C and its supplement, Figure 5—figure supplement 2 were shown in a new Supplementary figures (Figure 5—figure supplement 4, 5 and 6), showing no other non-specific bands.

*"Thus folding of the two main HEAT/HEAT-like domains forms an extensive internal cavity consistent with the shape that we observe in EM analysis, while providing an elegant explanation for the conundrum of how the polyglutamine tract located at the end of the N-terminal arm may affect change throughout the entire protein.": I fail to see this elegant explanation from Figure 7. They need a clearer model. Most HEAT importins show a super-helical structure with the internal face interacting with proteins to induce allosteric effects on the HEAT protein to modify the scaffold that are transduced along the scaffold (as described by Kleckner in one of the references). This has been done very well by Yuh-min Chook on the analysis of karyopherin Beta2, a huntingtin interactor and HEAT-rich protein.*

Elegant refers not to the complex interaction-data but only to the solution that the spherical shape of full-length huntingtin brings to the conundrum of how the amino terminal polyQ tract of full-length huntingtin may modulate the structural features of the full-length protein entire shape. As detailed in the response to Reviewer #1, Figure 4 was revised with the new data of Q46-huntingtin and Figure 7 was removed. The initial models are low-resolution but provide the first (initial) fundamental structural models of full-length huntingtin that are required to now carry out additional (future) comprehensive and higher-resolution studies of full-length huntingtin that will be needed to determine whether (or not) the predictions from the computational model detailed in Kleckner et al. are in detail found in full-length huntingtin folding and whether features of the high-resolution empirical-models for importin or karyopherin β2 are also observed in the very much larger full-length huntingtin protein or, alternatively whether the folding of each HEAT-domain protein is idiosyncratic due to the specific amino acid sequences of the HEAT-repeats and other features that distinguish each HEAT-repeat protein from another.

*Reviewer #3 (Additional data files and statistical comments): Need to see phospho-antibody validation data. This would include: dot blots to gauge affinity, full western blots to gauge specificity, as well as blots on extracts with either no or reduced huntingtin, and IF studies with antigen peptide competition.*

Since the manuscript is focusing on purified huntingtin structure and function, the current validation of our phospho-antibodies (see the detail in response above) would be enough to support our characterization of phosphorylated huntingtins with different sizes of polyglutamine tract.

[Editors’ note: the author responses to the re-review follow.]

*1) In Figure 1 the meaning of the dotted line is not explained. Please include one sentence for clarification as you did in the original version of the paper.*

Thank you for reminding us. We have included the sentence to explain the dotted line in the Figure 1 legend, as suggested.

*2) In the figure legend of Figure 1 exchange duplicated for duplicates.*

We have corrected it accordingly.

*3) From the Introduction: "Indeed, a comparison of lines of transgenic modified HTT BAC mice has implicated unique amino-terminal serine phosphorylation in mutant huntingtin gain of toxic function, indirectly implying a long-range impact of the amino-terminal region on huntingtin structure and function (Gu et al., 2009)." The Gu et al. manuscript clearly demonstrates the protective effect of serine 13 and 16 phospho-mimetic mutations in a Q84 context, and no effect of S13AS16A mutations. This is the opposite of this statement. Furthermore, another group showed protection in the YAC128 model by small molecules that induced this PTM.*

We agree that the work by Gu et al. has been wrongly quoted here and we have corrected the mistake in the Introduction.

*The proper reference for polyglutamine effects at distal regions of huntingtin is likely from Zerial's work on full-length huntingtin and HAP40 interactions at the carboxyl-terminus affected by the polyglutamine expansion. (Pal et al., J Cell Biol. 2006 Feb 13;172(4):605-18).*

This paper would better fit into the Discussion with other two papers that were also suggested to be considered for polyglutamine effects at distal regions of huntingtin and we have included them in the revised Discussion. Please see below.

*Another paper that deserves discussion in this contest is Schilling B, Gafni J, Torcassi C, Cong X, Row RH, LaFevre-Bernt MA, Cusack MP, Ratovitski T, Hirschhorn R, Ross CA, Gibson BW, Ellerby LM. J Biol Chem. 2006 Aug 18;281(33):23686-97. Epub 2006 Jun 16. This work mapped out phosphorylation sites across huntingtin, and showed a trend of all sites being hypo-phosphorylated due to polyglutamine expansion, even near the carboxyl-terminus.*

We have included this paper as well as two other papers that reported reduced phosphorylation levels due to polyglutamine expansion in the Discussion section.

*4) The authors claim in the Abstract that they provide the first glimpse into the structural properties of huntingtin and an elegant solution to the apparent conundrum of how the extreme amino-terminal polyglutamine tract confers a novel property on huntingtin, causing the disease. Some qualifiers or reformulation are indicated: this is not the first glimpse, a recent huntingtin structural paper demonstrated the α-solenoid structure of huntingtin. There are across a few labs many papers on huntingtin PTMs in full-length huntingtin and conclusions of reduced phosphorylation at well characterized sites. Many publications suggest a loss of function of mutant huntingtin in events post-development.*

We have removed “the first glimpse” term in our revised Abstract and changed the last sentence as: “Our work delineates the structural characteristics of full-length huntingtin, which are affected by the polyglutamine expansion, and provides an elegant solution to the apparent conundrum of how the extreme amino-terminal polyglutamine tract confers a novel property on huntingtin, causing the disease.”

Some are included above and below for the reference of the authors: Pal A, Severin F, Lommer B, Shevchenko A, Zerial M. J Cell Biol. 2006 Feb 13;172(4):605-18. Here they showed that interactions with HAP40 at the extreme c-terminus was influenced by the polyglutamine expansion on the carboxyl-terminus.

El-Daher MT, Hangen E, Bruyère J, Poizat G, Al-Ramahi I, Pardo R, Bourg N, Souquere S, Mayet C, Pierron G, Lévêque-Fort S, Botas J, Humbert S, Saudou F. EMBO J. 2015 Sep 2;34(17):2255-71. doi: 10.15252/embj.201490808. Epub 2015 Jul 12. This work showed that Amino-terminal proteolytic fragments could interact with Carboxy-terminal fragments of huntingtin, as well as toxicity by the carboxyl terminal fragments of huntingtin.

*Caron NS, Desmond CR, Xia J, Truant R. Proc Natl Acad Sci U S A. 2013 Sep 3;110(36):14610-5. doi: 10.1073/pnas.1301342110. Epub 2013 Jul 29. This work showed the folding back of huntingtin amino terminus to carboxyl-distal regions of huntingtin in both fragment and full-length endogenous huntingtin contexts.*

As suggested by the reviewers, we have discussed and included all the three references in the Discussion section.

*5) One of the referees questioned the data in Figure 3. S/he suggested "that the images were selected with the nanogold particles on the top in B (which are not as specific as gold-labelled antibodies), but it is difficult to gauge this relative to the rest of the structures, which all look very different. In my opinion, the data is not consistent with the 2D class average in 3D structures, and could just be the selection of images of a different orientation with the epitope up in the Z plane.*

*In 3A, I see obvious differences in the 2D-3D class average with or without the antibody in the top half of the structure."*

*Please address this concern in the new revision of the paper.*

The reviewer pointed out that the difference in 2D and 3D analysis clearly show the location of the antibody. As we agreed with the reviewers in that the nanogold particles does not further support the location of the amino-terminal region, we removed the nanogold particle images.

(Figure 3). Now the original Figure 3 is named as Figure 3.

*6) Some discussion of the structural data comparing with other HEAT-rich proteins would be beneficial. Most of these HEAT proteins have large, solvent exposed central cavity, this is very common in the nuclear transport field. So, in a large HEAT protein, this would be anticipated.*

We have discussed and compared the huntingtin structure and other HEAT-repeat protein inthe Discussion as “Other HEAT repeat proteins such as nuclear importin and exportins have functional protein-protein binding interface located at the inner side of the solenoid structure (Chook and Blober, Curr Opin Struct Biol 2001, Cingolani et al., Nature 1999). Considering that the size of huntingtin is much bigger that other HEAT repeat proteins, we can imagine that the HEAT repeat domains can be folded back to form a closed structure that we have observed in huntingtin, having functional sites located in the internal cavity.”

*7) Figure 5. Mab2166 is a good choice of antibody, as it is one of the few fully validated anti-human huntingtin antibodies. They need a loading control, or proper loading that has the Mab2166 signal within the linear range of the assays. This is a problem in 5C, the bands are too intense.*

This figure has been removed from the revised manuscript.

8) In Figure 6A, the first 2166 blots are also all darker than the range of the assay, and in the bottom blot comparing to pS2550, the levels of pS2550 clearly track with more or less total huntingtin in the 2166 blot (lanes 1 and 3). In both figures, they would be far more convincing with some additional cell biological imaging data in addition to the biochemical assays. This would give the reader more confidence.

This figure has been removed from the revised manuscript.